# Designing a norepinephrine optical tracer for imaging individual noradrenergic synapses and their activity in vivo

Matthew Dunn[1], Adam Henke[1], Samuel Clark [2,3,4], Yekaterina Kovalyova [1], Kimberly A. Kempadoo[5], Richard J. Karpowicz Jr.[1], Eric R. Kandel[3,5,6,7], David Sulzer[2,3,4] & Dalibor Sames[1]

Norepinephrine is a monoamine neurotransmitter with a wide repertoire of physiological roles in the peripheral and central nervous systems. There are, however, no experimental means to study functional properties of individual noradrenergic synapses in the brain. Development of new approaches for imaging synaptic neurotransmission is of fundamental importance to study specific synaptic changes that occur during learning, behavior, and pathological processes. Here, we introduce fluorescent false neurotransmitter 270 (FFN270), a fluorescent tracer of norepinephrine. As a fluorescent substrate of the norepinephrine and vesicular monoamine transporters, FFN270 labels noradrenergic neurons and their synaptic vesicles, and enables imaging synaptic vesicle content release from specific axonal sites in living rodents. Combining FFN270 imaging and optogenetic stimulation, we find heterogeneous release properties of noradrenergic synapses in the somatosensory cortex, including low and high releasing populations. Through systemic amphetamine administration, we observe rapid release of cortical noradrenergic vesicular content, providing insight into the drug's effect.

[1] Department of Chemistry, Columbia University, New York, NY 10027, USA. [2] Department of Neurology, Columbia University, New York, NY 10032, USA. [3] Department of Psychiatry, Columbia University, New York, NY 10032, USA. [4] Department of Pharmacology, Columbia University, New York, NY 10032, USA. [5] Department of Neuroscience, Columbia University, New York, NY 10032, USA. [6] Kavli Institute for Brain Science, New York, NY 10032, USA. [7] Howard Hughes Medical Institute, New York, NY 10032, USA. These authors contributed equally: Matthew Dunn, Adam Henke, Samuel Clark. Correspondence and requests for materials should be addressed to David Sulzer (email: ds43@columbia.edu) or to Dalibor Sames (email: ds584@columbia.edu)

Norepinephrine (NE: also known as noradrenaline) is a major neurotransmitter of the sympathetic peripheral nervous system that modulates the function of most visceral organs, glands, and the immune system[1]. NE is also a key neurotransmitter in the central nervous system (CNS). Brain NE neurons primarily originate in the locus coeruleus (LC)[2] and project axons throughout the CNS, including to the cortex, hippocampus, hypothalamus, amygdala, cerebellum, and spinal cord (Fig. 1c)[3–5]. Accordingly, the NE system plays numerous physiological roles, including brain-wide regulation of neurovascular coupling and cerebrospinal fluid flux[6], and local effects such as modulation of cortical and hippocampal neuronal circuitry[7].

While it is well established that the LC–NE system regulates stress responses, arousal, and sleep–wake cycles, more recent reports also suggest specific functions in cognition[7,8]. Shifts in LC neuronal firing between tonic and phasic activity control decision-making processes and facilitate optimization of behavior in a changing environment. In the "adaptive gain and behavior optimization hypothesis," LC neuronal firing controls the balance between exploitation of the ongoing task versus exploration of new ones[9]. NE synaptic inputs reset the activity states and firing patterns of ensembles of cortical neurons, presumably by modulating synaptic responses of these cells via activation of adrenergic receptors[7].

The NE system has also been implicated in numerous neurodegenerative and psychiatric disorders. For example, in Parkinson's disease, degeneration of LC–NE neurons often precedes loss of dopamine (DA) neurons in the substantia nigra[10]; in Alzheimer's disease, some of the most extensive loss of sub-cortical neurons occurs in the LC[11–13]. Modulators of NE neurotransmission represent an important class of widely used therapeutics, including adrenergic receptor agonists/antagonists, and modulators of NE storage, release, and reuptake. For example, the antidepressant mirtazapine is an α2 receptor blocker, and the antidepressant reboxetine is an inhibitor of the plasma membrane NE transporter (NET)[14,15]. Amphetamine, presently administered under the tradename Adderall, is used for treating attention-deficit hyperactivity disorder (ADHD)[15] in hundreds of thousands of children in the US[16], and is self-administered by millions worldwide[17]. Amphetamine enhances NE (and DA) transmission[18–20], although, whether the effects are due to NET inhibition or the redistribution of stored NE content (e.g., via reverse transport) at pharmacologically relevant doses has long been debated[21,22]. Despite the growing awareness of the importance of the LC–NE system, there have been no experimental tools to study NE release at the level of discrete synapses, and thus the release properties of the sparse cortical NE axons and synapses in vivo remain unknown.

The ongoing alteration of synaptic transmission provides a means for the organism's ability to learn and change behaviors[23]. Even individual synapses en passant on the same axon[24] are presumably regulated by ongoing expectation and experience to be more or less likely to participate in subsequent neurotransmission. To determine how behavior and sensory input alter specific synapses in vivo, novel tools to measure neurotransmission at individual synapses in vivo must be developed.

The current approaches for measuring NE lack the spatial and temporal resolution to discern individual synapses in the living brain. While electrochemical detection can distinguish DA from NE in some contexts[25–27], these approaches have insufficient spatial resolution to detect NE release from individual synapses[28], and are not useful in sparsely innervated regions of the CNS, such as the neocortex[29]. New fluorogenic cell-based sensors can detect changes in cortical NE in vivo with high sensitivity; these approaches, however, also lack the spatial resolution required to distinguish individual synapses[30,31]. Fluorogenic receptor-based sensors have been developed to selectively detect glutamate concentration changes in the brain with single synapse resolution[32–35], but similar sensors for monoamine neurotransmitters have not yet been reported. Chemical labels can be used for detection of NE with adequate spatial resolution, such as glyoxylic-acid-induced fluorescence histochemistry in fixed tissues[36,37], and more recently, fluorogenic chemosensors that react with NE in large dense core vesicles of cultured cells[38,39], but these techniques have not been adapted to living tissue.

Here we introduce the use of the fluorescent false neurotransmitter (FFN) concept to provide an optical tracer of NE neurotransmission[40]. Several generations of fluorescent DA mimetics have been developed in our laboratories, which revealed the heterogeneous nature of dopaminergic presynaptic activity in mouse brain tissue[41,42]. FFN102 is an exemplar dopaminergic FFN (Fig. 1), deriving its labeling selectivity for dopaminergic neurons as a substrate of the plasma membrane DA transporter (DAT)[41,43]. As shown below, however, FFN102 is not a substrate for the NET, and thus a poor optical tracer for NE.

We now expand the scope of the FFN concept by introducing the first NE-FFN, probe FFN270 (Fig. 1). This compound is designed as a fluorescent NET substrate and also a substrate of the neuronal vesicular monoamine transporter (VMAT2), the transporter responsible for packaging neurotransmitter in monoaminergic synaptic vesicles. The probe is taken up along with NE into synaptic vesicles within NE axonal varicosities and enables measurement of synaptic vesicle content release by destaining during exocytosis. FFN270 enables both an examination of noradrenergic microanatomy and an observation of synaptic activity in the cortex of intact neuronal circuits in vivo. The use of FFN270 also demonstrated that exposure of animals to pharmacologically relevant concentrations of amphetamine exerts a dramatic effect on cortical NE axons, providing new insights to the drug's mechanism of action.

## Results

**Design, synthesis, and photophysical properties of NE-FFN.** We approached the global design of NE-FFN probes by incorporating the key structural feature of NE, the arylethylamine fragment, into the coumarin fluorophore (Fig. 1a). We chose this design as the FFN probe must act as a substrate of both NET on the plasma membrane of NE neurons and VMAT2 on the NE synaptic vesicles. Selective binding of these compounds to their protein targets is not sufficient, as they need to be actively transported in a manner similar to the native neurotransmitter across the plasma and vesicular membrane via ion gradient-coupled processes. Thus, a major portion of the structural space of molecular fluorophores was not applicable, due to their large size. The relatively small coumarins, however, proved to be excellent fluorophores for this purpose, due to their high brightness-to-size ratio, photostability, and biocompatibility. Aided by the relative substrate permissiveness of NET[19,44,45], we pursued the development of coumarin-based NET substrates.

Lead identification was initiated by examining the DA-FFNs, FFN102 and FFN202, which have been successfully used to study dopaminergic projections in the striatum of acute murine brain slices[41]. To test the applicability of FFNs in vivo, the compounds were applied (50 μM, 30 min) to the frontal cortex of mice through exposed cranial windows. Whereas FFN102 showed no appreciable uptake above background, we observed numerous punctate axonal strands with FFN202 (Supplementary Fig. 2). Since FFN102 did not label these same axons, we hypothesized that they did not express DAT, but instead the closely related NET, and thus represented NE axons. Consistently, we observed a small level of accumulation of FFN202 in human embryonic

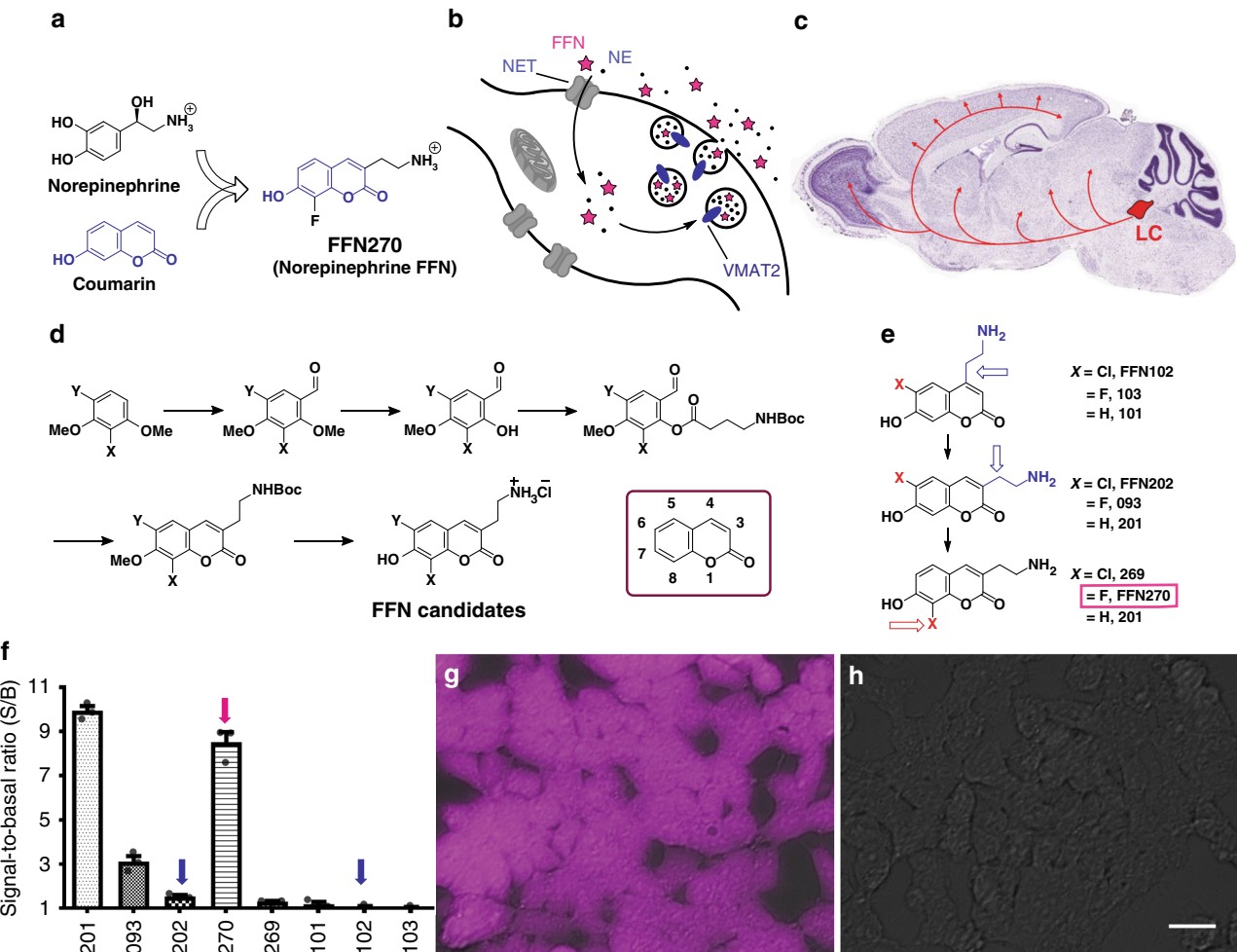

**Fig. 1** Design of NE-FFNs. **a** The design of NE-FFNs combines structural features of NE with the coumarin fluorescent core. **b** NE-FFNs trace NE uptake from the extracellular space, packaging into vesicles, and exocytosis as they are designed to be substrates of NET and VMAT2. **c** Representative illustration of NE neuron distribution in the brain (Allen Institute)[51]. Many NE neurons originate from the locus coeruleus (LC) and project to the majority of brain regions. **d** General synthetic scheme for preparation of 3-series aminoethyl-7-hydroxycoumarins as potential NE-FFN candidates. See Methods for experimental conditions. **e** Focused series of candidate NE-FFNs. **f** Total cellular fluorescence after loading of FFN candidates (5 μM) in hNET-HEK cells. Signal-to-Basal ratio (S/B ± SEM) was determined by comparing FFN fluorescence in the presence and absence of 2 μM nomifensine (NET inhibitor) after a 30 min incubation period. Blue arrows highlight previously described DA-FFNs, and a magenta arrow highlights FFN270, the leading pH-sensitive NE-FFN candidate (S/B: 8.5 ± 0.4, mean ± SEM, $n = 3$ independent experiments run in triplicate). Representative images of FFN270 without inhibitor (**g**) and with inhibitor (**h**). Scale bar: 20 μm

kidney cells stably transfected with human NET (hNET-HEK) (Fig. 1f). Testing compound **201** showed that the 3-aminoethyl coumarin core of FFN202 was a far better NET substrate (Fig. 1e and f). Despite its favorable signal, however, **201** is not a well-suited fluorescent probe for brain tissue, due to its insufficient brightness and $pK_a$ of the phenolic group ($pK_a = 8.0$), which is outside of the desired range to distinguish probe localization in secretory and synaptic vesicles (pH ~5.5) versus the extracellular space (pH ~7.4). Both limitations can be addressed by introducing an electron-withdrawing group in the adjacent position to the coumarin phenol, thus providing the rationale for the development of a series of related, halogen-substituted coumarins.

Synthesis of a library of fluorophores with desirable photophysical properties and phenol $pK_a$ values was accomplished with a 5-step synthetic sequence using 2- and 4-haloresorcinols as commercially available starting materials (Fig. 1d). The presence of an electron-withdrawing chlorine or fluorine atom in the 6- and 8- position (Fig. 1e) increased acidity of the phenolic group ($pK_a = 5.9$–$6.3$) and shifted the $pK_a$ to the desired physiologically

relevant pH range (5.5–7.4, Supplementary Table 1). At the cytosolic pH of ~7.4, the haloanalogs are mostly in a phenolate form, which is considerably brighter than the protonated phenol form. As a result of this acidification, all newly prepared FFN candidates bearing a halogen atom displayed greater brightness than FFN201 ($pK_a = 8.0$). These compounds, including FFN270, exhibit two resolved absorption/excitation maxima depending on solvent pH (FFN270 ex: 320 nm or 365 nm, em: 475 nm, Supplementary Fig. 1) and thus can function as ratiometric fluorescent pH-sensors.

## Development of fluorescent coumarin substrates of NET. The
collection of 7-hydroxycoumarin compounds (Fig. 1e) was examined for possible NET substrates using a multi-well fluorometric assay to measure NET-dependent uptake in hNET-HEK cells. The cells were grown in 96-well plates and incubated with the experimental compounds (5 μM, 30 min) in the absence and presence of the NET inhibitor, nomifensine (2 μM). The activity of the compounds at NET was determined by comparing the

fluorescence intensity between the uninhibited wells (signal S; NET-dependent and independent/nonspecific uptake) and the inhibited wells (basal signal B; only NET-independent/non-specific uptake) and expressed as a signal-to-basal ratio (S/B) (Fig. 1f).

The 4-series hydroxycoumarin fluorophores (bearing the aminoethyl group in the 4-position of coumarin), including the dopaminergic tracer FFN102[41,43], showed insignificant activity at NET; whereas **201**, the parent compound of 3-series regioisomers, showed robust transporter-dependent fluorescence accumulation (~10-fold S/B, Fig. 1). Introducing the fluorine atom in position 6 of the 3-series coumarin ring, affording compound **093**, improved the photophysical properties, but negatively affected the ability of the fluorophore to act as a substrate for NET. Increasing size of the halogen atom in this position caused a further decrease in NET activity (H (**201**) > F (**093**) > Cl (**FFN202**)), with FFN202 showing only weak NET uptake (<2-fold S/B). In contrast, placing the fluorine atom in position 8, generating FFN270, led to a small reduction in specific uptake (8.5 ± 0.4; mean ± SEM) compared to the lead compound **201** (9.9 ± 0.4, $n = 3$, $P = 0.04$, two-tailed unpaired $t$-test, Fig. 1f), while still improving photophysical properties ($pK_a = 6.25$). However, additional increase in size of the halogen atom at position 8 eliminated NET activity (**269**, <2-fold S/B).

To confirm the hNET substrate activity of FFN270 demonstrated in the fluorometric study, we used epifluorescence microscopy to visually compare cellular uptake and labeling morphology of uninhibited and inhibited hNET-HEK cells using nomifensine (2 μM) or cocaine (1 μM), as well as control HEK293 cells. In hNET-transfected cells, FFN270 exhibited a homogeneous staining pattern indicative of cytosolic distribution (Fig. 1g), while only negligible uptake was found in nomifensine treated (Fig. 1h), cocaine treated (Supplementary Fig. 3), and null-transfected cells (Supplementary Fig. 3), confirming the NET-dependent uptake identified from the initial screen. We also examined hDAT and hSERT-transfected HEK cells, and found that FFN270 is selective for hNET (Supplementary Fig. 4).

**FFN270 is also a substrate of VMAT2.** To determine the potential of FFN270 for loading NE synaptic vesicles and trace NE neurotransmission, we examined VMAT2-dependent transport using a protocol employing HEK cells stably transfected with rat VMAT2 (VMAT2-HEK). This system, where VMAT2 is expressed and active on acidic intracellular organelles of HEK cells, has been employed extensively by our laboratories[43,46,47]. VMAT2 activity was evaluated using wide-field epifluorescence microscopy, which enables visualization of individual cells and the intracellular distribution of the fluorescent compounds (Fig. 2c–f). In contrast to NET-HEK cells, these cells do not over-express NET on the plasma membrane, and so higher concentrations of the probe (20 μM) and longer incubation times (2 h) were used to facilitate passive diffusion.

VMAT2-HEK cells that were treated with FFN270 displayed a bright punctate fluorescent pattern consistent with accumulation of the compound in acidic organelles expressing VMAT2 (Fig. 2c).

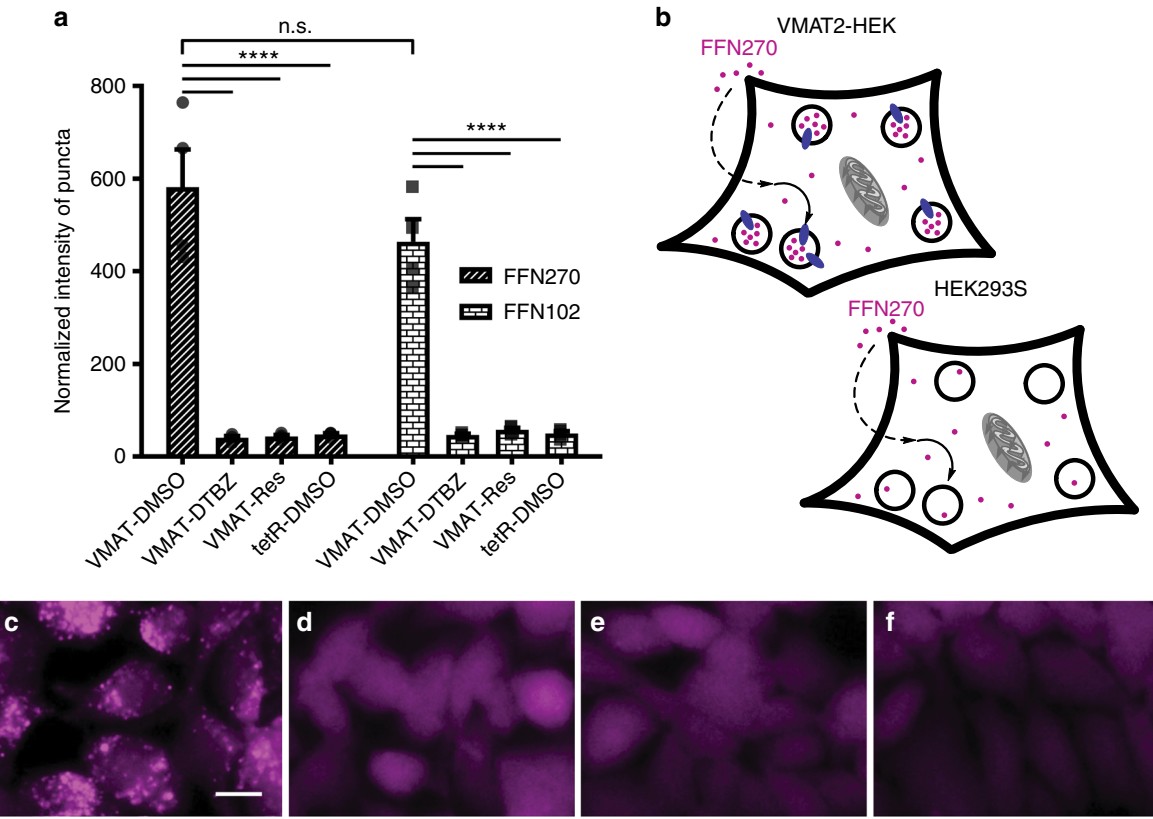

**Fig. 2** FFN270 is a VMAT2 substrate. **a** VMAT2-dependent loading of FFN102 and FFN270 were compared using the normalized intensity of puncta (number of puncta multiplied by average fluorescence intensity ± SEM, $n = 4$ independent experiments) in VMAT2-HEK cells after a 2 h incubation (20 μM) under different conditions. FFN270 showed robust puncta intensity (581 ± 81), comparable to FFN102 (464 ± 49), that was significantly diminished in dTBZ (41 ± 4) and reserpine (44 ± 3) inhibited conditions ($P < 0.0001$, one-way ANOVA Dunnett test). **b** Representative schematic depicting the rationale for intracellular punctate fluorescence with active VMAT2, and general cytosolic labeling in null-transfected or inhibited conditions. The probes passively diffuse through the plasma membrane and are then actively concentrated in acidic compartments that express VMAT2. Representative images of FFN270 in **c** VMAT2-HEK cells, **d** dTBZ (2 μM) or **e** reserpine (2 μM) inhibited VMAT2-HEK cells, and **f** null-transfected HEK293 cells. Scale bar: 10 μm

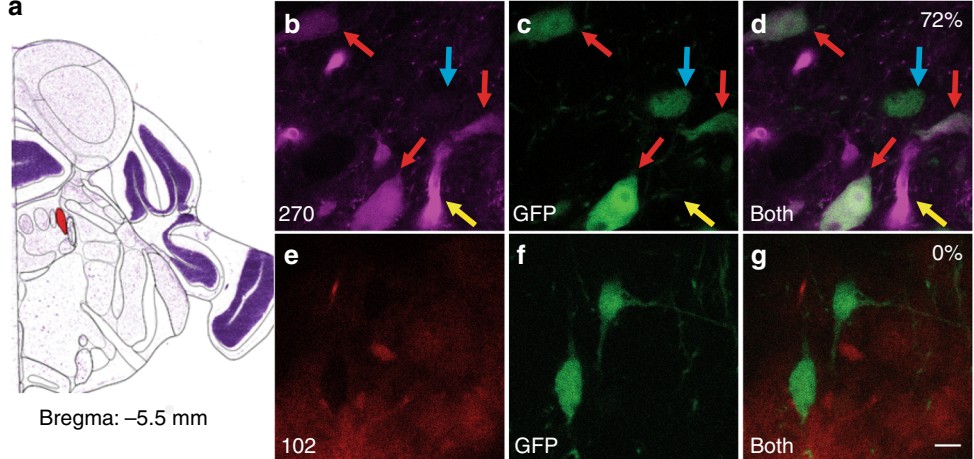

**Fig. 3** FFN270 labels NE neurons in the locus coeruleus. **a** Atlas image highlighting in red the location of the locus coeruleus in the mouse brain (Bregma: −5.5 mm, Allen Institute)[51]. **b–d** FFN270 (magenta, **b**) effectively colocalized with the noradrenergic label TH-GFP (green, **c**), resulting in a 72% colocalization of noradrenergic cell bodies (labeled by red arrows, 62/86 cells, 6 animals). Cells that did not colocalize are highlighted with a blue arrow, and blood vessels are highlighted with a yellow arrow. **e–g** When repeated with FFN102 (red, **e**), no colocalization was observed with TH-GFP (**f**). Images obtained by 2-photon microscopy in acute mouse brain slice (for imaging parameters, see Methods). Scale bar: 10 μm

This VMAT2-dependent accumulation was expressed as normalized total puncta fluorescence (number of puncta multiplied by average puncta fluorescence intensity, Fig. 2a). In contrast, only general cytosolic fluorescence was observed in the null-transfected HEK cell line (not expressing VMAT2, Fig. 2f) or in VMAT2-HEK cells pretreated with two different VMAT inhibitors, dTBZ (2 μM, Fig. 2d) or reserpine (2 μM, Fig. 2e). Comparing the VMAT2-uptake of FFN270 with FFN102, a previously confirmed VMAT2 substrate used to measure DA vesicular exocytosis ex vivo[43], confirmed that FFN270 accumulated in acidic organelles in a VMAT2-dependent manner with similarly normalized intensity ($n = 4$, $P = 0.26$, one-way ANOVA; Dunnett's test, Fig. 2a), indicating that this probe is a dual NET-VMAT2 substrate.

**FFN270 does not bind to monoamine and other CNS targets**. A primary screening assay was conducted with FFN270 against 54 CNS molecular targets, including the monoamine receptors and transporters (in collaboration with the Psychoactive Drug Screening Program, University of North Carolina at Chapel Hill)[48]. The screen showed no significant binding of FFN270 to any of the receptors examined (a positive hit was defined as >50% inhibition by the experimental ligand at 10 μM, Supplementary Table 2). As with any compound used in pharmacological studies, there remain other potential targets including orphan receptors that cannot be excluded, but this primary assay indicated a clean pharmacological profile for FFN270, an important prerequisite for the FFN probe design.

**FFN270 labels NE neuronal soma and axons ex vivo**. After reaching the in vitro criteria of an NE-FFN—including dual transporter substrate activity (NET and VMAT2), favorable photophysical properties (sufficient brightness and phenol p$K_a$ within vesicle physiological pH range), a clean profile for the studied receptors, and low basal cellular labeling—we examined whether FFN270 is suitable for labeling neurons by applying the probe to the LC region of acute mouse brain slices (Fig. 3), the primary region for noradrenergic cell bodies in the brain. Using a mouse line expressing green fluorescent protein (GFP) under the tyrosine hydroxylase promoter (TH-GFP)[49], GFP is expressed in catecholamine neurons, and can be used as a reference marker for noradrenergic neurons in the LC, as this region does not contain

dopaminergic cell bodies[50]. We observed that 72% of noradrenergic cells expressing GFP were also labeled with FFN270 (greater than $2 \times$ SD of the background, 62/86 cells, 6 animals, Fig. 3b–d), while previous generation DA-FFN probes, such as FFN102, showed no measurable uptake by any GFP-labeled cells in this region (Fig. 3e–g).

We next determined the degree to which FFN270 labels the projections of these noradrenergic cell bodies by measuring uptake in the outer layers (Layer 1–3) of the barrel cortex (Fig. 4a)[51]. This brain region receives extensive noradrenergic innervation[2], and NE-dependent neuromodulation has been shown to affect the strength and temporal patterns of sensory input-activated barrel cortical networks[52–54]. After FFN270 incubation in acute brain slice, we observed significant labeling of noradrenergic axons (Fig. 4b), as well as larger structures that we confirmed were blood vessels by lectin staining (Supplementary Fig. 8). Due to the relatively sparse density of labeled projections, the representative 2D images shown are maximum Z-projections of 20-μm-thick volumes. Consistent with observations in the LC, FFN102 did not label these cortical noradrenergic projections (Fig. 4c), even at very high concentrations (50 μM) (Supplementary Fig. 9). As predicted by preliminary in vivo results and the hNET-HEK assay, FFN202 and **093** showed low and moderate levels of uptake, respectively (Supplementary Fig. 5). FFN270 loading of these structures was inhibited by nomifensine (2 μM), or the selective NET inhibitor reboxetine (500 nM, Fig. 4d, e), indicating that NET-dependent transport was required for FFN270 uptake into these projections. The inhibition of uptake was quantified by comparing the number of puncta in the presence of nomifensine ($7.7 \pm 0.7\%$, $P < 0.001$) or reboxetine ($17.5 \pm 3.5\%$, $P < 0.01$, $n = 3$, average of 2 slices per condition from 3 different animals, one-way ANOVA; Dunnett's test, Fig. 4f) to untreated slices. However, the uptake in blood vessels was unchanged for both inhibition conditions.

We further confirmed that the FFN270-labeled projections in the barrel cortex were noradrenergic using TH-GFP animals as a fluorescent marker[49]. We observed significant axonal colocalization between GFP and FFN270 (88.9%, 154/176 axons, 4 animals, Fig. 4g–i). We quantified colocalization by measuring the fraction of GFP-labeled axons that also contained significant FFN labeling (>$2 \times$ SD of background). In contrast to the LC, however, there is potentially some dopaminergic innervation in this area[55], and

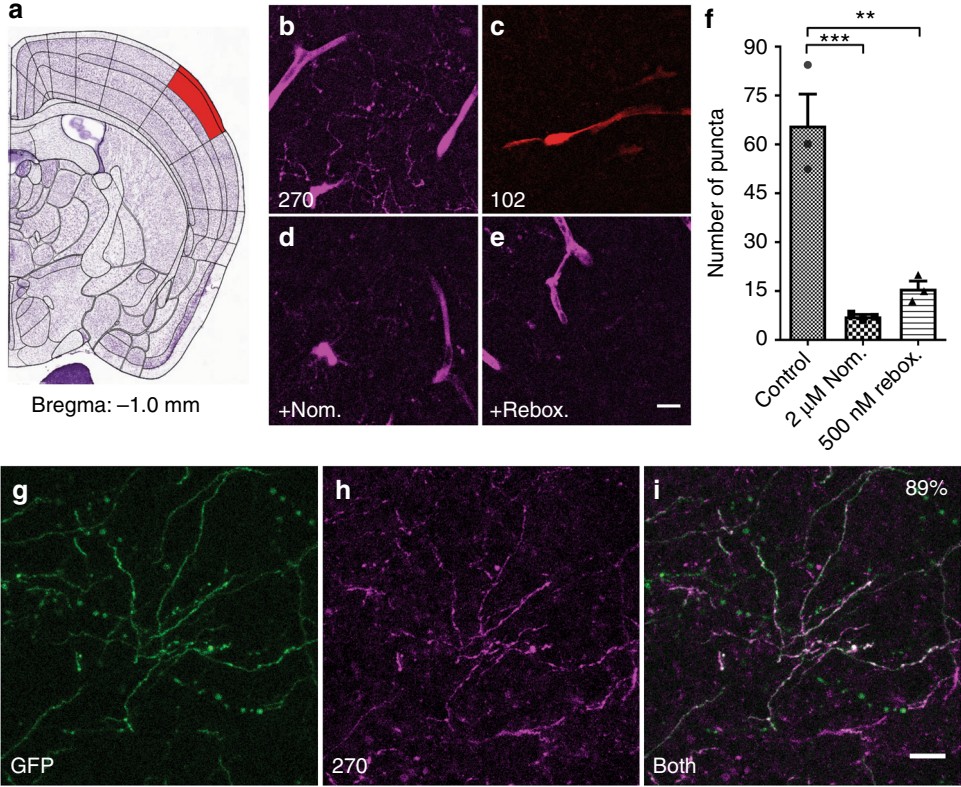

**Fig. 4** FFN270 is accumulated in NE axons of the barrel cortex. Representative images FFN270 (**b**) and FFN102 (**c**) loaded into layer 1 of the barrel cortex (**a** Bregma: −1.0 mm, Allen Institute)[51] of acute murine brain slices (10 μM, 30 min, 20 μm Z-projections, 2-photon microscopy images). Noradrenergic axons appeared as long strings with punctate release sites, while blood vessels appeared as wider tube structures. FFN270 axonal labeling in Layer 1 of the barrel cortex was inhibited by nomifensine (**d** 2 μM, Nom.) or reboxetine (**e** 500 nM, Rebox.). **f** The average number of puncta per region of interest was significantly higher in control conditions (68.5 ± 22) than with Nom. (7.7 ± 0.7, $P < 0.001$) and Rebox. (17.5 ± 3.5, $P < 0.01$, one-way ANOVA; Dunnett's test, $n = 3$, average of 2 slices per condition from 3 different animals). Blood vessel labeling was not inhibited. **g–i** FFN270 (**g**) labeling also highly colocalized (88.9%, 154/176 axons, 4 animals) with TH-GFP signal (**h**). All data listed as mean ± SEM. Scale bar: 10 μm

GFP in this transgenic mouse line does not distinguish between dopaminergic and noradrenergic projections. We therefore examined colocalization in TH-Cre animals that were injected only in the LC with a virus that expresses the fluorescent marker, enhanced yellow fluorescent protein (EYFP) in the presence of Cre recombinase (AAV/2/5.EF1a.DIO.hChR2(H134R)-EYFP. WPRE.hGH, Supplementary Fig. 6)[56]. In these animals, we observed substantial EYFP expression in cortical noradrenergic projections of the barrel cortex, and a high degree of colocalization between the fluorescent marker and FFN270 (72.4%, 92/127, 6 animals, Fig. 5c–e). The fraction of FFN projections that did not contain EYFP is consistent with our TH-GFP colocalization when accounting for the efficiency of viral transduction (90–96%)[56], and indicates that fewer than 10% of FFN270-labeled axons could be dopaminergic. This conclusion is further supported by an absence of fluorescent reporter found in this region in a mouse line expressing tdTomato under a DAT-Cre driven promoter (see Methods), as well as the lack of substantial FFN270 uptake in a brain region rich in DAT expression, the dorsal striatum (Supplementry Fig. 9).

**FFN270 reveals functional heterogeneity of NE release sites.** We next examined whether the punctate fluorescence in the noradrenergic axons represented the uptake and storage of FFN270 in synaptic vesicles, and whether FFN270 could be released using electrical and channelrhodopsin-mediated stimulation techniques. Using electrical stimulation (10 Hz, 3000 pulses), we observed destaining and non-destaining varicosities, and in the former population a single exponential decay in background-subtracted puncta fluorescence over time with a half-life of 34.6 s, a significant increase in FFN release when compared to the non-stimulated control slices (Figs. 5a, 2–3 slices per animal, 3 animals). The rate of FFN270 release from active puncta was comparable to the release rate observed with other FFNs in dopaminergic synapses[42]. Repeating the electrical stimulation while inhibiting calcium channels using $Cd^{2+}$ (200 μM) led to an 87% reduction in the number of identified destaining puncta (2 out of 143 puncta versus 17 out of 155), confirming that FFN270 release was due to calcium-dependent stimulated exocytosis (Supplementary Fig. 12, 2–3 slices per animal, 3–4 animals).

Using a complementary optogenetic system in which mice unilaterally express channelrhodopsin-2 (ChR2) and EYFP in noradrenergic neurons (viral delivery as described above), we also measured activity-dependent FFN270 vesicular release triggered by 470 nm light stimulation (10 Hz, 2400 pulses). As local ChR2 activation selectively excites noradrenergic axons, these results exclude potential local circuit effects that may result from electrical stimulation of the acute brain slice. Since the FFN emission wavelength overlaps with the stimulation light, we could not monitor FFN fluorescence during the stimuli, and therefore measured fluorescence before and after the stimuli. We compared the change in fluorescence of individual FFN-labeled axons that colocalized with the ChR2-EYFP signal (ChR$^+$) to axons that did not express ChR2-EYFP (ChR$^-$). In the absence of ChR, there was a baseline 27.6 ± 1.3% loss in fluorescence after 4 min of

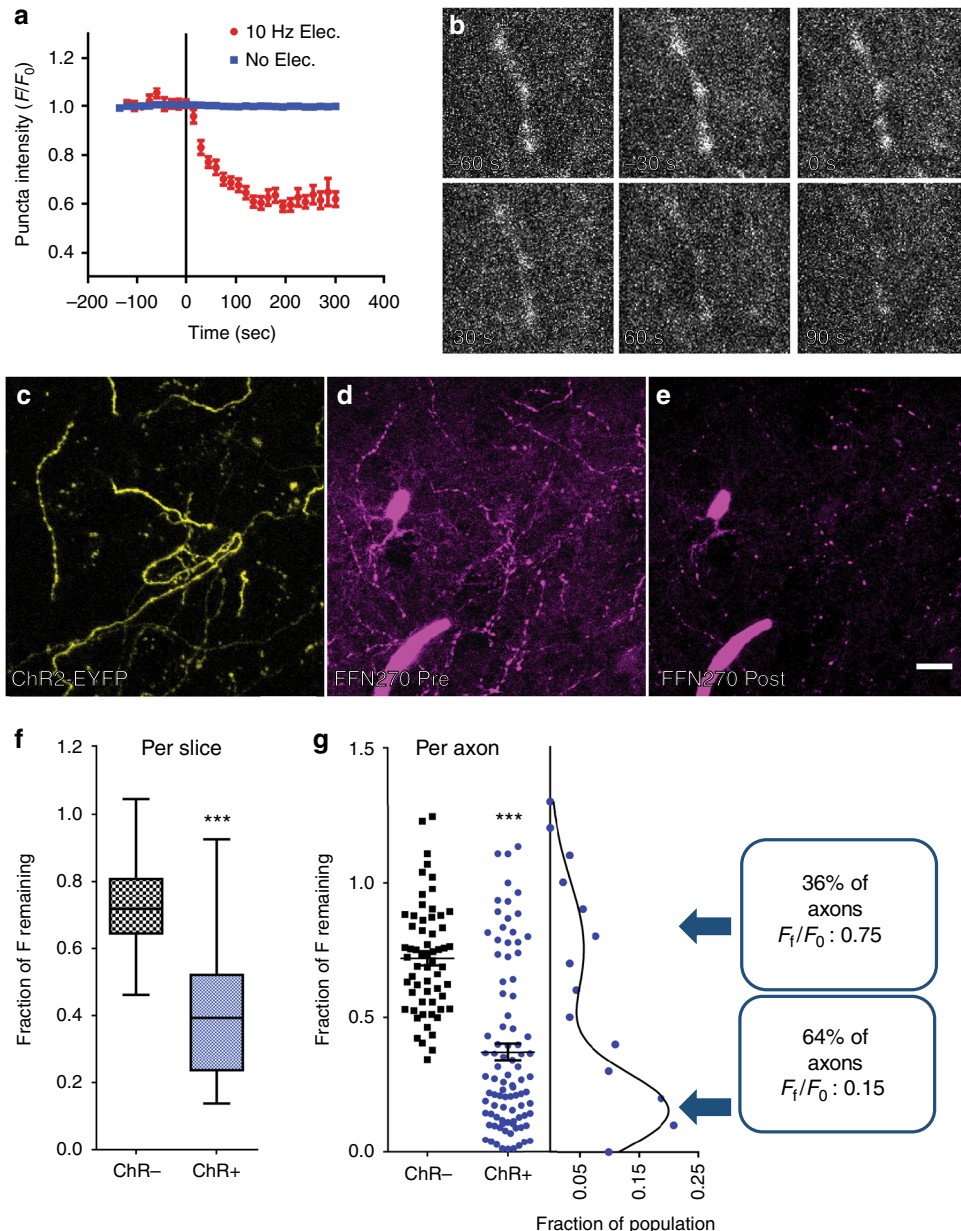

**Fig. 5** Axonal FFN270 release reveals two populations of NE synapses. **a** Change in FFN270 puncta fluorescence in the barrel cortex of acute slice during the course of a locally applied electrical stimulation (10 Hz, 3000 pulses, starts at t: 0 min). In red are destaining puncta identified during electrical stimulation (t-$_{1/2}$: 34.6 s), and in blue are puncta during no stimulation ($n = 2$–3 slices per animal, 3 animals for each condition). **b** A representative FFN 270 labeled noradrenergic axon during electrical stimulation. FFN270 (**d**) strongly colocalized with the ChR2-EYFP signal (**c** 72.4%, 92/127 axons, 6 animals) in TH-Cre animals injected with floxed-ChR2-EYFP in the LC. The ChR2 was then locally stimulated with 470 nm light (10 Hz, 2400 pulses), and the change in FFN270 signal after stimulation (**e**) was measured. **f** When averaged across each slice, there was a significant decrease in remaining FFN270 signal that colocalized with ChR2-EYFP (ChR$^+$, 42.6 ± 5.9%) compared to those that did not (ChR$^-$, 72.4 ± 3.3%, $P < 0.0001$, two-tailed unpaired $t$-test, $n = 2$–3 slices per animal from 5 animals, data presented with min/max whiskers). **g** We also compared fluorescence remaining of the individual axons across all trials, and observed a similar trend (ChR$^+$: 37.1 ± 3.3%, ChR$^-$: 72.0 ± 2.6%, $P < 0.0001$, Mann–Whitney). The individual axons that comprise the ChR$^-$ population follow a normal distribution (D'Agostino & Pearson test, $P = 0.3$), while the ChR$^+$ population does not ($P = 0.003$). The ChR$^+$ population closely follows a double Gaussian distribution (right panel histogram, $R^2 = 0.91$) with high (64% with 15.1% F remaining) and low releasing (36% with 75.4% F remaining) populations. All data listed as mean ± SEM. Scale bar: 10 μm

stimulation. In puncta with ChR, the loss in fluorescence was 57.4 ± 7.9%, a 2.1-fold increase (2–3 slices per animal, 5 animals, $P < 0.0001$, two-tailed unpaired $t$-test, Fig. 5c–f).

We also compared the FFN release between individual NE axons within the ChR$^+$ and ChR$^-$ populations. FFN destaining from the noradrenergic axon population of the ChR$^-$ group fit a normal distribution (D'Agostino & Pearson test, $P = 0.3$), but the ChR$^+$ group did not ($P = 0.003$, Fig. 5g). As reported with

dopaminergic FFNs[42], by tracing neurotransmitter release at single synapse resolution, a trend of low releasing or "silent" presynaptic sites and a separate high releasing population was apparent. The fluorescent changes of the ChR$^+$ group were well fit by the sum of two normal distributions with distinct peaks ($R^2 = 0.91$). While some of the ChR2$^+$ axons (34%) only released 24.6 ± 2.2% of their fluorescent content, the majority of axons (66%) almost fully destained, releasing 84.9 ± 10.9% of their

content (Fig. 5g). We confirmed this distribution response was similar for the entire group of acute brain slices by conducting a normality test on the same ChR$^+$ axons grouped by slice, which indicated a normal distribution (*P* = 0.2).

We next examined the effect of stimulus frequency by measuring the amount of FFN released at 1 Hz (240 pulses) and 10 Hz (2400 pulses), using transgenic optogenetic tool delivery by crossing the TH-Cre line with one expressing ChR2 controlled by a loxP site (Ai32, see Methods). In the transgenic line, we measured significant release of FFN270 from ChR2$^+$ axons at both 10 Hz (loss of fluorescence, 65.4 ± 3.0%) and 1 Hz (55.5 ± 3.0%, Supplementary Fig. 10, 2 slices per frequency per animal, 4 animals): these results were not statistically different from each other (*P* = 0.07, Mann–Whitney), or from the 10 Hz condition using viral expression (62.9 ± 3.2%, *P* = 0.05, Kruskal–Wallis). Consistent with previous findings, the 10 Hz stimuli did not produce a normal distribution of puncta destaining (D'Agostino and Pearson test, *p* < 0.0001), although the 1 Hz stimulation condition was better fit (*P* = 0.2).

To examine the axons with remaining FFN signal post-electrical or optogenetic stimulation, we then exposed slices to amphetamine (10 μM), which produced a nearly complete loss of puncta after 5 min (90.8 ± 3.2%, 2–3 slices per animal, 3 animals, Supplementary Fig. 7), suggesting that the remaining fluorescence in these "silent" boutons was located in healthy noradrenergic axons that express functional NET and/or VMAT2 and can undergo amphetamine-induced reverse transport. The remaining 9.2% of puncta that did not destain after the entire time course likely represent a nonspecific signal of FFN270 in acute brain slices, and was comparable to the number of puncta that remained in slice after inhibition with nomifensine (7.2 ± 0.7%, Fig. 4f) and did not colocalize with the TH-GFP signal (11.1%, Fig. 4i). As a control, perfusion of the NET inhibitor (nomifensine, 2 μM) on a slice preloaded with FFN270 had no effect on the rate of FFN signal decay compared to ACSF controls (Supplementary Fig. 7, one-way ANOVA Dunnett test, *P* = 0.95, *n* = 2 slices per animal, 3 animals), suggesting very low spontaneous FFN reverse transport and reuptake in the acute brain slice. This control supports a model in which amphetamine-dependent release is due to a redistribution of catecholamines from synaptic vesicles to the cytosol from where it undergoes reverse transport[57,58], and that in this system, inhibition of FFN reuptake plays little role in amphetamine effects.

Together, the results in acute mouse brain slices confirm that FFN270 actively accumulates into noradrenergic synaptic vesicles and provides the first probe capable of quantifying noradrenergic synaptic release with single synapse resolution in brain tissue, and a first insight into functional properties of NE synapses in the CNS.

**FFN270 is an optical tracer of NE in vivo**. We next used two-photon imaging of the superficial cortical layers through a cranial window to characterize the potential of FFN270 to study noradrenergic axons in the brain of living animals. For in vivo experiments, an imaging window in the skull over the barrel cortex was created, and FFN270 (100 pmol) was injected through the window into the anesthetized animal at depths from 20 to 100 μm from the surface of the brain. In contrast to our original in vivo FFN lead, FFN202, we observed a dramatically increased number of FFN-labeled axons using FFN270, with a better signal-to-background-fluorescence ratio that persisted for over 2 h (Fig. 6).

We found that FFN270 could be used to study evoked NE release in vivo using the virally expressed ChR2 system described

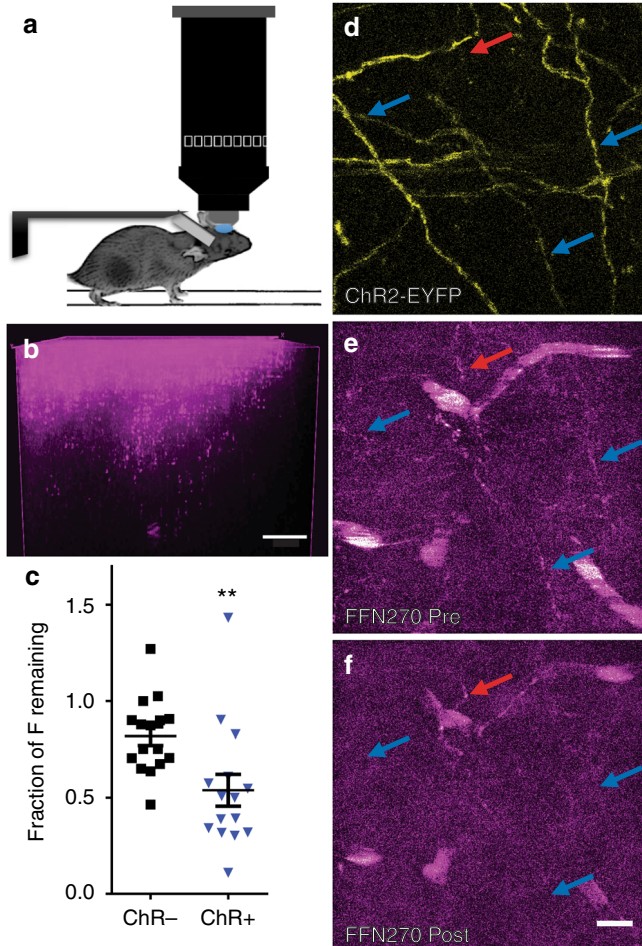

**Fig. 6** Examining NE axons and release sites in living animals. **a** Representative in vivo setup using an anesthetized head-fixed animal to image Layer 1 of the barrel cortex. **b** A representative 3D reconstruction of FFN270 labeling (50 μM locally applied) in Layer 1 of the barrel cortex in vivo. Scale Bar: 20 μm. **c** FFN270 loaded into NE axons with ChR2, as described in Fig. 5, can be released with local 470 nm light stimulation (10 Hz, 2400 pulses). There is a significant increase in FFN270 released from axons that colocalize with ChR2 (ChR$^+$, 46.2 ± 8.3%, highlighted with blue arrows) compared to axons that do not (ChR$^-$, 18.13 ± 4.8%, highlighted with a red arrow, *n* = 6 animals, *P* = 0.0012, Mann–Whitney). Representative images of the ChR2-YFP signal (**d**), and the FFN270 signal before (**e**) and after (**f**) optogenetic stimulation. All data listed as mean ± SEM. Scale bar: 10 μm

above (Fig. 6d–f). We measured the change in FFN fluorescence of loaded axons before and after 470 nm light stimulation (10 Hz, 2400 pulses), and compared the changes observed in axons that did (ChR$^+$) and did not (ChR$^-$) colocalize with the ChR2-EYFP signal. We observed a 2.5-fold increase in the amount of released FFN from ChR$^+$ axons (46.2 ± 8.3%) compared to ChR$^-$ axons (18.1 ± 4.8%, mean ± SEM, *n* = 6 different animals, *P* = 0.0012, Mann–Whitney, Fig. 6c). With FFN270, we could clearly distinguish the differences in firing rates between noradrenergic axons undergoing the slow tonic firing observed in anesthetized animals[59], and those that were optogenetically stimulated. In the living animal, we also observed differences in the distribution of FFN release from individual axons with and without ChR2. The ChR$^-$ population was well fit by a normal distribution (D'Agostino & Pearson, *p* = 0.38) while the ChR$^+$ population was not (*P* = 0.002), suggesting that NE axons possess different

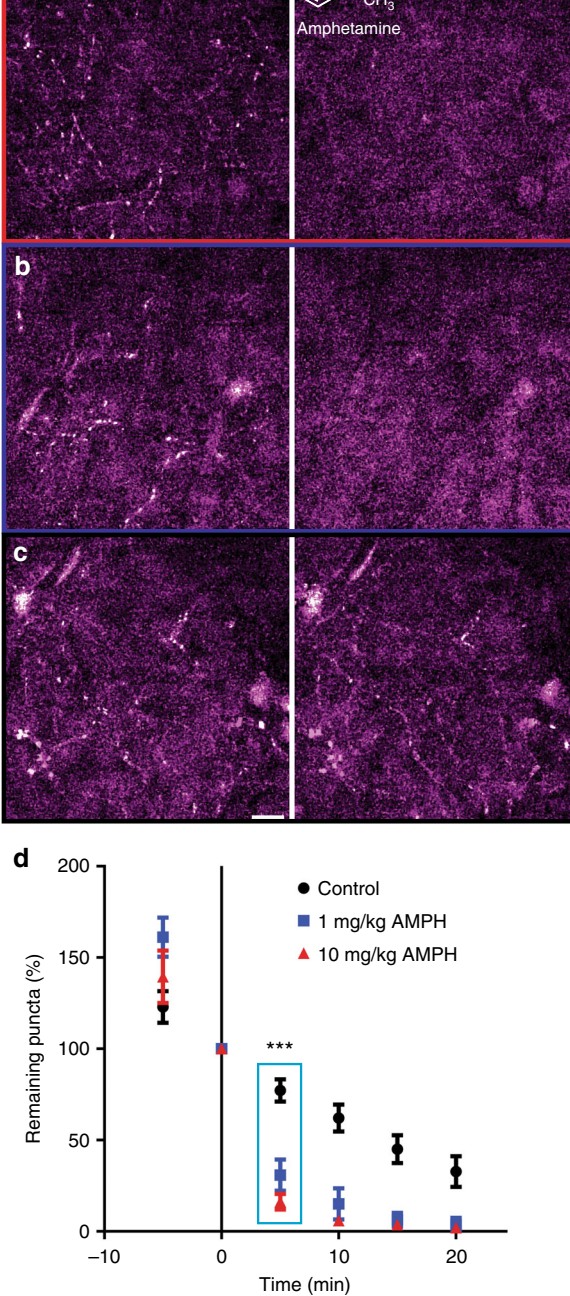

**Fig. 7** Amphetamine releases FFN270 from NE projections in vivo. Representative images of FFN270 in Layer 1 of the barrel cortex in living mice taken before and 5 min after injection of amphetamine (AMPH). The left column includes images acquired before i.p. administration of a high dose (10 mg/kg, **a**), low dose (1 mg/kg, **b**) and a vehicle (saline only, **c**), while the right column is post-injection. **d** Quantification of the number of puncta before and after each AMPH condition over a 25 min time course. After 5 min, there was already a significant decrease in remaining FFN270 puncta in the high (83.9 ± 4.4%, $n = 5$) and low (69.1 ± 8.6%, $n = 6$) AMPH condition when compared to the control (22.8 ± 6.1%, $n = 7$, $P < 0.0003$ for both AMPH conditions, one-way ANOVA Dunnett test). All data listed as mean ± SEM. Scale bar: 10 μm

release properties in vivo, consistent with the findings ex vivo. In contrast to results in the brain slice, however, we were unable to fit the in vivo ChR$^+$ axons to a double Gaussian with well-defined low and high releasing populations.

We then used FFN270 to examine the effect of acute amphetamine treatment on the noradrenergic system of a living mammal. While amphetamine is currently used to treat ADHD in hundreds of thousands of patients and is abused by millions of people, there is a debate on how amphetamine treatment affects individual noradrenergic projections in the cortex[60,61]. Using the same in vivo FFN270 loading and imaging technique, we measured changes in FFN content during two amphetamine treatments (1 and 10 mg/kg) administered through intraperitoneal injection (i.p.)[62]. Over 25 min, we observed significantly more FFN released in each amphetamine condition compared to saline-injected controls (Fig. 7). After 5 min, we observed decreases of 69.1 ± 6.1% (1 mg/kg, $n = 6$ different animals) and 83.9 ± 4.4% (10 mg/kg, $n = 5$ different animals) in the number of FFN-positive structures after amphetamine treatment, compared to 22.8 ± 6.1% for saline-injected controls ($n = 7$ different animals, $P < 0.0003$ for both amphetamine treatments, one-way ANOVA Dunnett test, Fig. 7d). After 15 min, amphetamine-induced FFN270 destaining was nearly complete, with almost total release from all FFN270 loaded noradrenergic projections (>95% reduction in puncta).

## Discussion

The introduction of FFN270, the first FFN developed to label noradrenergic synapses, has to our knowledge, provided the first tool to image the primary function of synaptic transmission—synaptic vesicle exocytosis—in the CNS in vivo. Coupled with electrical and optogenetic stimulation, we were able to examine exocytosis-dependent release from NE axonal branches and individual NE release sites in murine brain tissue and within the intact cortical circuitry of the living brain. The results revealed heterogeneity of neurotransmitter release and identified a population of sites that release only a small portion of vesicular content, even after long trains of photonic pulses delivered locally to the cortical NE axons. This finding is in accordance with our recent study using FFN200, which demonstrated that a portion of dopaminergic puncta in the striatum do not release significant amounts of FFN200 (and thus DA) and are therefore functionally "silent", and is consistent with a new study indicating that only a fraction of DA synapses possesses presynaptic scaffolding proteins that enable synaptic vesicle fusion[63]. In addition to the introduction of optical detection of neurotransmitter release in vivo, the present study differs from previous studies with FFNs in multiple ways: (1) we examined ensembles of single NE release sites, which originate from a distinct neurochemical cell type (LC–NE rather than dopaminergic neurons); (2) the examined NE release sites are in the barrel cortex (in contrast to DA sites in the dorsal striatum); (3) the NE axon stimulation was achieved by selective targeting of ChR2 and local photostimulation (in contrast to local current injection). The presence of synaptic heterogeneity in two distinct neuronal populations, in two separate brain regions, and under various stimulation methods, suggests that this may be a general phenomenon and raises multiple questions: What mechanisms underlie differences in presynaptic activity? Can the silent sites be activated during behavior? Do the silent sites serve as reserve pools of synaptic vesicles? These and other fundamental questions can now be addressed in specific circuits using the FFN probes and associated imaging methods. Alternative stimulation techniques, such as red-shifted ChRs, will further enable more detailed examination of release properties of NE release sites by measuring FFN release during optogenetic stimulation of NE axons (local stimulation) or somata (distal stimulation).

We used FFN270 in vivo to address whether systemic amphetamine treatment affects release from individual cortical

noradrenergic projections, previously not possible with standard microdialysis or electrochemical techniques. These experiments show that pharmacologically relevant levels of amphetamine (e.g., 1 mg/kg, i.p.), that produce behavioral effects[64], can drive FFN release from LC axons in the cortex in vivo. While we examined an NE-like compound rather than NE itself, this approach indicates that amphetamine acts as an NE releaser and not simply as a NET blocker, a question that has been controversial in elucidating the therapeutic effects of this drug in ADHD[22,65]. As amphetamine-induced neurotransmitter redistribution is mostly thought to act via reverse transport across uptake transporters rather than synaptic vesicle exocytosis[58,66], and all FFN270-labeled axonal varicosities were uniformly destained in the cortex both in vivo and ex vivo, our findings indicate that even the low releasing sites contain neurotransmitter accumulation transporters and competent storage vesicles.

FFN270 addresses the shortcomings of, and complements, the existing methods used for NE detection in the brain[30,31,36–39]. As a chemically targeted molecular agent, FFN270 should be applicable to primates and other species not amenable to genetic manipulation[67,68]. However, the probe does not detect the neurotransmitter itself and cannot measure the vesicle content of NE versus DA (NE is synthesized from DA in NE neurons)[56] or the changes of NE concentrations at or near the release sites. We envision that the combination of FFNs and cell-based fluorogenic NE sensors[30,31] or protein-based sensors will provide powerful experimental tools for detailed synaptic-level examination of NE release and neurotransmission. In broad terms, multiplexing FFNs with optical techniques that measure receptor activation by neurotransmitters promises a means to distinguish the modulation of neurotransmission at discrete synapses, a process fundamental to learning and behavior.

## Methods

**Synthesis and preparation of NE-FFN leads**. Detailed synthetic routes (Supplementary Fig. 13, 14) and confirmation of compound identities is included in Supplementary Methods.

**Photophysical characterization and measurement of log D**. Photon absorption and emission spectra were measured using a BioTek H1MF plate reader on bottom read mode. UV absorption spectra were acquired by adding probe (4 μL of 10 mM stock solution in DMSO) to 996 μL of PBS buffer at different pH values (final probe conc. = 40 μM) in a clear-bottom white 96-well plate. Excitation/emission spectra were taken by adding probe (5 μL of 40 μM solution in PBS) to 995 μL of PBS buffer of different pH values (final probe conc. = 0.2 μM) in a clear-bottom white 96-well plate. The p$K_a$ values of probes were determined from the absorption spectra. The absorbance $\lambda_{abs, max}$ of the red-shifted peak was plotted versus pH of the PBS solution; the data were fit to a sigmoid curve using GraphPad Prism 5 software to determine the p$K_a$ value. The log $D$ values were determined by a traditional shake flask method. Each measurement was performed in duplicate as follows. First, 20 μM probe solution in 1 mL PBS (pH 7.4) was prepared to which 1 mL of $n$-octanol was added and mixed thoroughly. The mixture was kept in dark for three days for complete equilibrium, and the concentrations of probe in each layer were determined based on the UV absorbance. Log $D$ values were determined based on the following equation; log $D$ = log[probe]$_{oct}$ – log[probe]$_{PBS}$, where [probe]$_{oct}$ and [probe]$_{PBS}$ are the concentrations of the probe in $n$-octanol and PBS, respectively.

**Characterization of probe uptake in transfected cell culture**. For these experiments, stably transfected HEK cells were used to determine specific transporter uptake, and citations are included for the previous validation of this system. The stably hNET-transfected HEK cells[69] were seeded at a density of 0.08–0.09 × 10$^6$ cells/well in white solid-bottom 96-well plates and allowed to proliferate in growth medium (DMEM + 10% FBS + 100 U/mL Penicillin and 100 μg/mL Streptomycin) for ~2 days at 37 °C to reach confluence. On the day of the experiment, the complete growth medium was aspirated, wells were washed with 200 μL PBS, and treated with 100 μL experimental medium (phenol-red-free DMEM + 1% FBS + 100 U/mL Penicillin and 100 μg/mL Streptomycin) with DMSO (vehicle, 0.02% v/v) or nomifensine (2 μM). The cells were incubated for 60 min at 37 °C, and experimental compounds (100 μL/well of 10 μM solution in experimental medium with DMSO (vehicle) or nomifensine 2 μM) was added for a final concentration of 5 μM compound in wells. Cells were then incubated for

30 min at 37 °C. The experiment was terminated by two rapid PBS washes (200 μL/well) followed by addition of fresh PBS buffer (120 μL/well). The fluorescence uptake in cells was immediately recorded using a BioTek H1MF plate reader (3 × 3 area scan, bottom read mode) with excitation and emission wavelengths set at 370 nm and 460 nm, respectively. Substrate activity at hNET was determined using signal-to-basal ratio (S/B): mean fluorescence uptake (with DMSO vehicle) divided by that in the presence of nomifensine. Data are presented as normalized uptake ± SEM from three independent experiments (eight separate measurements per condition per experiment).

For epifluorescence microscopy of probes in hNET-HEK cells and their respective controls, HEK293, cells were plated onto poly-D-lysine (Sigma Aldrich, conc. = 0.1 mg/mL) coated clear-bottom six-well plates at a density of 0.15–0.20 × 10$^6$ cells per well and grown at 37 °C in 5% CO2. Following ~4 days of growth, the cells had reached 80–90% confluence. The culture medium was removed by aspiration, and the cells were washed with PBS (Invitrogen, 1.0 mL/well). To investigate the inhibitory effects of nomifensine maleate salt (Sigma-Aldrich) and cocaine hydrochloride salt (Sigma-Aldrich) on probe uptake, cells were incubated in 0.9 mL of experimental media containing the inhibitor (2 μM prepared from 10 mM stock solution in DMSO) or DMSO vehicle as a control at 37 °C in 5% CO2 for 1 h. The compound uptake was initiated by adding 0.1 mL of experimental media containing probe (200 μM, prepared from a 10 mM stock solution in DMSO, with a final probe concentration of 20 μM in the uptake assay) with and without nomifensine (2 μM) or cocaine (1 μM). Following incubation at 37 °C in 5% CO2 for 30 min, the media was removed by aspiration and the cells were washed with PBS (1 mL/well) and maintained in fresh experimental media (1 mL/well). Fluorescence images (at least three images/well in duplicate wells) were acquired with a Leica FW 4000 imaging system (Leica Microsystems) equipped with a Chroma custom filter cube (ex = 350 ± 25 nm, em = 460 ± 25 nm; Chroma Technology Corporation) and a Leica DFC-360FX camera. Fluorescence and bright field images were acquired with exposure time set at 600 ms and 37 ms, respectively. All images were adjusted using the same contrast and brightness level using ImageJ software (National Institutes of Health).

For epifluorescence microscopy of probes in VMAT-HEK cells[46,47] and their respective controls, TetR-HEK, cells were plated at a density of 0.15–0.20 × 10$^6$ cells per well on poly-D-lysine coated 6-well optical plates (Falcon) and grown at 37 °C in 5% CO2. Following ~4 days of growth, the cells had reached 80–90% confluence. The culture medium was removed by aspiration, the cells were washed with PBS (Invitrogen, 1.0 mL/well), and the wells were pretreated with experimental medium with or without the VMAT2 inhibitors reserpine (1 μM, Sigma-Aldrich) or dihydrotetrabenazine (DTBZ, 2 μM; Sigma-Aldrich) for ~60 minutes. To initiate uptake, solutions of experimental probe with or without inhibitor were added to the appropriate wells for a final concentration of 20 μM probe with or without reserpine (1 μM) or DTBZ (2 μM). Cells were incubated at 37 °C for 2 h, at which point the probe solutions were removed by aspiration and wells were gently washed with PBS (2 mL). Wells were maintained in fresh experimental medium and were imaged as described above. VMAT2 substrate activity was quantified using the Multiple Thresholds ImageJ plug-in (created by Damon Poburko, Simon Fraser University, Burnaby, BC, Canada) was used for quantification of fluorescent puncta in images of VMAT2-HEK cells obtained from fluorescence microscopy. Puncta were identified as objects conforming to defined parameters: brightness (>0.6 SDs above background), appropriate size (0.5–4 μm$^2$), rounded shape (min circularity: 0.6), and well delimited boundaries. Data presented as mean intensity of these puncta structures per image, normalized to uptake by tetR-HEK cells as background ± SEM from four independent experiments (six images per condition per experiment).

**FFN loading and imaging in acute murine brain slices**. All wild-type animals used for slice preparation were C57BL/6 mice obtained from the Jackson Laboratory (Bar Harbor, ME), and include both males and females. For slices containing the LC, animals were sacrificed at 20 days due to peak NET expression at an early age[70]. For all other experiments mice were used at 7–9 weeks. All animal protocols were approved by the IACUC of Columbia University. Mice were decapitated and acute 300-μm-thick coronal slices were cut on a Leica VT1200 vibratome (Leica Microsystems) at 4 °C and allowed to recover for 1 h in oxygenated (95% O$_2$, 5% CO$_2$) artificial cerebrospinal fluid (ACSF) containing (in mM): 125 NaCl, 2.5 KCl, 26 NaHCO$_3$, 0.3 KH$_2$PO$_4$, 2.4 CaCl$_2$, 1.3 MgSO$_4$, 0.8 NaH$_2$PO$_4$, 10 Glucose (pH 7.2–7.4, 292–296 mOsm/l). Slices were then used at room temperature for all imaging experiments.

Slices were incubated with the FFN (10 μM) for a 30 min bath incubation and then transferred to an imaging chamber (QE-1, Warner Instruments, Hamden, CT) and held in place by platinum wire and nylon string custom made holder and superfused (1 ml/min) with oxygenated ACSF[71]. Slices were allowed to wash in the perfusion chamber for at least 10 min before imaging. For any inhibition experiments, slices were first pre-incubated with the inhibitor (1 μM nomifensine or 500 nM reboxetine) for 15 min, and then co-incubated with the FFN for the 30 min loading period. Inhibitor was also added to the perfusing ACSF solution. Fluorescent structures were visualized at depths of at least 25 μm from the slice surface using a Prairie Ultima Multiphoton Microscopy System (Prairie Technologies, Middleton, WI) with a titanium-sapphire Chameleon Ultra II laser (Coherent) equipped with a 60 × 0.9 NA water immersion objective. FFN270 was

excited at 760 nm and 440–500 nm light was collected. Images were captured in 16-bit $112 \times 112$ µm² field of view at $512 \times 512$ pixel² resolution and a dwell time of 10 µs/pixel using Prairie View software. For imaging the sparse noradrenergic cortical projections 20-µm-thick z-stacks (1 µm/image) were collected and then compressed using a maximum z-projection. For Supplementary Fig. 9, the concentration of FFNs was increased to 50 µM.

**Colocalization of FFN270 and CNS markers**. For colocalization with blood vessels, acute murine brain slices were loaded with FFN270 as described above. Low zoom 2-photon images (16-bit, $350 \times 350$ µm² of view at $1024 \times 1024$ pixel²) were collected with a $10 \times 0.25$ NA air objective using all other previously described excitation/emission parameters. After imaging, the slices were then fixed for >48 h using 4% paraformaldehyde in PBS and then immunostaining was performed as previously described[72]. Slices were then washed with 0.3% Triton X-100 PBS (PBS-T) for 1 h and then incubated with biotinylated solanum tuberosum lectin (20 µg/mL, Vector, Lot No. ZE0-2C-0319) overnight at 4 °C. After three washes with PBS-T, streptavidin/DyLight-488 conjugate (Thermo Scientific) was incubated at 20 µg/mL for 2 h at room temperature. Slices were then rinsed extensively with PBS and then imaged using the same 2-photon parameters. DyLight 488 was excited with 950 nm light and emission was collected between 490 and 560 nm.

Colocalization of FFN270 and noradrenergic structures was performed in animals that express GFP under the tyrosine hydroxylase promoter (TH-GFP)[73]. Acute slices from the LC or barrel cortex were incubated with FFN270 as described above and imaged using interlaced scanning with two separate lasers. FFN was imaged using 740 nm excitation and an emission range of 435–485 nm, and GFP was imaged using 920 nm excitation and an emission range of 500–550 nm. To quantify the number of GFP-positive cells in the LC that colocalize with FFN, the cells were manually selected if they contained fluorescence in either channel that was greater than two standard deviations above background, and reported as the percentage of GFP-positive cells that also contain FFN270 signal. Quantification of the colocalization of noradrenergic projections in the barrel cortex was determined by identifying FFN270 puncta using the Multiple Thresholds plug-in for the ImageJ software. Any ROIs that corresponded to fluorescence in blood vessels were manually discarded. The identified ROIs were then considered to positively colocalize if greater than 50% of the pixels overlapped with GFP signal that was greater than two standard deviations above background, and reported as the number of visibly discernable GFP axons with FFN270-postive puncta. For colocalization with dopaminergic structures, DAT-IRES-Cre mice[+/−] (Jackson Laboratory) were crossed with Ai9(RCL-tdT) mice[+/+] (Jackson Laboratory). Acute slices were collected from the dorsal striatum and barrel cortex. FFN was imaged as described above, and tdTomato was excited with 740 nm light and 570–640 nm emission light was collected.

**FFN270 release in acute murine brain slices**. For better temporal resolution (15 s/frame), imaging during electrical stimulation experiments used a smaller $54 \times 54$ µm² area and 5-µm-thick z-stack. At the start of electrical stimulation, a 10 Hz stimulation train (6 min total, each pulse 200 ms × 180–200 mA) was applied locally to the barrel cortex by a bipolar stainless steel electrode using an ISO-Flex stimulus isolator triggered by a Master-8 pulse generator (AMPI, Jerusalem, Israel). Quantification of fluorescent changes in destaining puncta was performed using a custom made Matlab script[42] available from our laboratory's website at http://sulzerlab.org/. Briefly, this script corrects shifts in the x,y, and z plane, subtracts background fluorescence, then picks puncta, and measures fluorescence of each over time. The kinetics of release were then determined from the fluorescent traces using GraphPad Prism 5 by fitting with a single exponential decay function.

To stimulate with optogenetics, channelrhodopsin-2 (ChR2) was selectively expressed in noradrenergic neurons using TH-IRES-Cre[+/−] (Jackson Laboratory) animals and a Cre-Lox adenoassociated virus (AAV/2/5.EF1a.DIO.hChR2 (H134R)-EYFP.WPRE.hGH, obtained from Penn Vector) injected into the LC (AP: −5.45, ML: + 1.30, DV: −3.80) unilaterally as described by our laboratories previously[56]. Animals were then allowed to recover for 4 wk prior to use in FFN experiments. Alternatively, ChR2 was expressed by crossing the TH-IRES-Cre[+/−] line with the Ai32(RCL-ChR2(H134R)/EYFP)[+/+] line (Jackson Laboratory). To measure colocalization in the barrel cortex, FFN270 was imaged using 760 nm excitation and an emission range of 435–485 nm, and EYFP was imaged using 950 nm excitation and an emission range of 500–550 nm. Quantification of colocalization between EYFP-positive axons and FFN270 puncta was conducted as described above and reported as percentage of FFN axons that also contain EYFP signal greater than two standard deviations above background. Stimulation of ChR2 was achieved using a 470 nm LED source (LED4D067 and DC4100 driver, Thorlabs) to provide blue light pulses (10 Hz, 5 ms duration, 2400 pulses total or 1 Hz, 5 ms duration, 240 pulses total) locally to the brain slice. Due to the overlap between FFN270 emission and the ChR2 excitation source, changes in fluorescence were measured only before and after stimulation. Z-stacks were maximal projected and then the before and after image stacks were registered using the Correct 3D Drift plug-in[74]. For quantification, images were background subtracted using a rolling ball with a 10 µm diameter, and then puncta from the pre-stimulation stack were selected using the Multiple Threshold plug-in. The mean fluorescence from puncta of a visually continuous axon were then compared to the mean fluorescence post-stimulation. Each FFN270-positive axon was then grouped based on whether

there was colocalization with EYFP signal as determined by signal greater than two standard deviations above background. Slices from both the injected and non-injected brain hemispheres were used for analysis.

To measure changes in FFN270 following amphetamine (AMPH) or nomifensine treatment, acute brain slices containing the barrel cortex were prepared and loaded with FFN270 as described above. Twenty micrometre thick z-stacks were imaged every 60 s for 15 min total. AMPH (10 µM) or nomifensine (2 µM) was added to the ACSF and then perfused over the slice starting at 5 min. These z-stacks were then registered using the Correct 3D Drift plug-in and the AMPH-dependent fluorescent changes were quantified by both the number of puncta selected by the Multiple Threshold plug-in and the average fluorescent intensity of those puncta over time.

**FFN270 loading and release in vivo**. Mice were anesthetized with isoflurane (1–4%). The depth of anesthesia was monitored by both the animal's response to toe-pinch every 5 min and observation of respiratory rate. Once the proper depth of anesthesia was reached, the animal was placed in a small animal stereotactic apparatus on top of a heating pad and Puralube vet ointment was applied to the eyes to prevent vision loss. All surgical procedures were performed in a sterile manner. Marcaine (0.5%) with saline 1:2 (0.25%) (2 mg/kg), was injected sub-cutaneously along the midline of the scalp as a local anesthetic. Before the incision, the hair over the scalp area was removed with NAIR and sterilized with a gentle scrub with Betadine on a sterile cotton swab, followed by 70% ethanol, repeated three times. The scalp was then removed with surgical scissors, membranes over the skull were removed by scraping, and the exposed bone was dried with compressed air. A 3 mm cranial imaging window was then made over the barrel cortex with a high speed dental drill (Midwest Stylus 360). The meninges were inspected to make sure there was no damage. Any bleeding from the bone was stopped with an application of collagen foam (Avitene Ultrafoam, Bard Davol). A glass pipette attached to a Nanoject II (Drummond Scientific) was filled with a 1 mM solution of FFN270 diluted in ACSF and attached to the stereotax. A volume of 100 nL of FFN270 was injected into the brain at coordinates AP: −0.9 mm to −1.0 mm and ML: 3.0 mm, over several depths (DV: 100 µm, 50 µm, and 20 µm) with 3 min of delay per depth. The glass pipette was then withdrawn from the brain and a plastic ring was glued around the window with Loctite 454 to hold ACSF for imaging. A metal headmount was then glued to the nearby exposed bone with Loctite 454. The mouse was then injected with ketamine (100 mg/kg) and xylazine (10 mg/kg) i.p. and then weaned off of isoflurane while making sure to maintain depth of anesthesia. The mouse was then head-fixed under the multiphoton microscope for imaging. The body temperature of the mouse was maintained for the duration of imaging with a heating pad.

For optogeneic stimulation, the ChR2 animal preparation, blue light stimulation parameters, and analysis were conducted as described above for acute brain slice experiments. The ChR2-positive animals that were used for ChR2 colocalization and FFN270 release in brain slice were also used for the in vivo experiments. Due to a larger degree of z-shift that was present in vivo, the imaging parameters for this experiment were modified slightly to collect from a larger z-stack (30 µm, 1 µm/image), while all other imaging parameters remained the same. The noradrenergic projections of the barrel cortex were then locally stimulated 35 min post-FFN270 injection (waited 30 min after initial injection to allow NET-dependent accumulation and natural clearance of unloaded FFN270 probe from background tissue, followed by two pre-stimulation image stacks). For these experiments, cranial windows were only created on the same hemisphere as the viral injection, and the data from both the ChR2-positive and -negative populations came from the same hemisphere.

For amphetamine (AMPH) experiments, after FFN270 was loaded, 30 µm z-stacks were collected every 5 min starting from 30 min post-FFN270 injection. After two image stacks, AMPH (0, 1, or 10 mg/kg) in Dulbecco's phosphate-buffered saline was injected into the intraperitoneal cavity. Images stacks were collected for 30 min post-injection. These stacks were registered using the Correct 3D Drift plug-in, and then puncta were selected using the Multiple Threshold plug-in using ImageJ. For the z-planes that remained consistent across the entire time course, the number of identified puncta were compared for each time point and normalized to the z-stack collected just before AMPH injection.

**Statistics**. Data is presented as mean ± SEM and standard parametric tests were used for statistical analysis. For the comparison of two different conditions a two-tailed $t$-test was used (Mann–Whitney if populations were not a normal distribution), while a one-way ANOVA followed by a Dunnett's multiple comparisons post-test was used for comparing conditions of three or more (Kruskal–Wallis if populations were not a normal distribution). The sample size was chosen based on similar studies in the field and based on our previous studies, the exact sample size for each result is incorporated in the text. To determine the normality of a particular distribution we used the D'Agostino & Pearson test. Data collection and analysis were not randomized or performed blind to the conditions of the experiments.

**Data availability**. Data sets and raw images are available from the corresponding author upon reasonable request.

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

# ARTICLE

56. Kempadoo, K. A., Mosharov, E. V., Choi, S. J., Sulzer, D. & Kandel, E. R. Dopamine release from the locus coeruleus to the dorsal hippocampus promotes spatial learning and memory. *Proc. Natl Acad. Sci. USA* **113**, 14835–14840 (2016).

57. Freyberg, Z. et al. Mechanisms of amphetamine action illuminated through optical monitoring of dopamine synaptic vesicles in Drosophila brain. *Nat. Commun.* **7**, 10652 (2016).

58. Sulzer, D. How addictive drugs disrupt presynaptic dopamine neurotransmission. *Neuron* **69**, 628–649 (2011).

59. Aston-Jones, G. & Bloom, F. E. Activity of norepinephrine-containing locus coeruleus neurons in behaving rats anticipates fluctuations in the sleep-waking cycle. *J. Neurosci.* **1**, 876–886 (1981).

60. Berridge, C. W. & Stalnaker, T. A. Relationship between low-dose amphetamine-induced arousal and extracellular norepinephrine and dopamine levels within prefrontal cortex. *Synapse* **46**, 140–149 (2002).

61. Blanc, G. et al. Blockade of prefronto-cortical α1-adrenergic receptors prevents locomotor hyperactivity induced by subcortical D-amphetamine injection. *Eur. J. Neurosci.* **6**, 293–298 (1994).

62. Salahpour, A. et al. Increased amphetamine-induced hyperactivity and reward in mice overexpressing the dopamine transporter. *Proc. Natl Acad. Sci. USA* **105**, 4405–4410 (2008).

63. Liu, C., Kershberg, L., Wang, J., Schneeberger, S. & Kaeser, P. S. Dopamine secretion is mediated by sparse active zone-like release sites. *Cell* **172**, 706–718.e15 (2018).

64. Daberkow, D. P. et al. Amphetamine paradoxically augments exocytotic dopamine release and phasic dopamine signals. *J. Neurosci.* **33**, 452–463 (2013).

65. Heal, D. J., Smith, S. L., Gosden, J. & Nutt, D. J. Amphetamine, past and present-a pharmacological and clinical perspective. *J. Psychopharmacol.* **27**, 479–496 (2013).

66. Sulzer, D., Sonders, M. S., Poulsen, N. W. & Galli, A. Mechanisms of neurotransmitter release by amphetamines: a review. *Prog. Neurobiol.* **75**, 406–433 (2005).

67. Wakayama, S. et al. Chemical labelling for visualizing native AMPA receptors in live neurons. *Nat. Commun.* **8**, 14850 (2017).

68. Er, J. C. et al. NeuO: a fluorescent chemical probe for live neuron labeling. *Angew. Chem.* **54**, 2442–2446 (2015).

69. Jensen, N. H. et al. N-Desalkylquetiapine, a potent norepinephrine reuptake inhibitor and partial 5-HT1A agonist, as a putative mediator of quetiapine's antidepressant activity. *Neuropsychopharmacology* **33**, 2303–2312 (2008).

70. Sanders, J. D., Happe, H. K., Bylund, D. B. & Murrin, L. C. Development of the norepinephrine transporter in the rat CNS. *Neuroscience* **130**, 107–117 (2005).

71. Wong, M. Y., Sulzer, D. & Bamford, N. S. Imaging presynaptic exocytosis in corticostriatal slices. *Methods Mol. Biol.* **793**, 363–376 (2011).

72. Bucher, E. S. et al. Medullary norepinephrine neurons modulate local oxygen concentrations in the bed nucleus of the stria terminalis. *J. Cereb. Blood. Flow. Metab.* **34**, 1128–1137 (2014).

73. Parslow, A., Cardona, A. & Bryson-Richardson, R. J. Sample drift correction following 4D confocal time-lapse imaging. *J. Vis. Exp.* **86**, e51086 (2014).

74. Parslow, A., Cardona, A. & Bryson-Richardson, R. J. Sample drift correction following 4D confocal time-lapse imaging. *J. Vis. Exp.* https://doi.org/10.3791/51086 (2014).

## Acknowledgements

This work was funded by the G. Harold & Leila Y. Mathers Charitable Foundation (to D. Sames), the National Institute of Mental Health, NIMH (R01MH108186, to D. Sames and D. Sulzer, NIDA07418 (D. Sulzer), and 1F31MH109293-01A1, to S. Clark), the JPB Foundation (D. Sulzer), the Burroughs Wellcome Fund Postdoctoral Enrichment Program, (to K. Kempadoo), and the Howard Hughes Medical Institute (to E. R. Kandel). D. Sulzer is a NARSAD Brain and Behavior Distinguished Investigator. We thank Dr. Wenbiao Gan (NYU) for assistance with exploratory in vivo imaging experiments, Dr. Mark Wightman for suggesting the use of lectin labeling of blood vessels, and Dr. Gary Aston-Jones for valuable discussions on the noradrenergic system. We also thank Dr. Rob Edwards (UCSF) for the VMAT2-HEK cells and Dr. Bryan Roth (UNC at Chapel Hill) for the NET-HEK cells.

## Author contributions

M.D. and D. Sames initiated the project. D. Sames and D. Sulzer directed the project. D. Sames and A.H. designed the compounds. A.H. synthesized the library of derivative candidates, examined candidates in transfected cell cultures, and characterized their photophysical properties with assistance from Y.K. and R.J.K. M.D. characterized uptake and release parameters of lead candidates in acute murine brain slice. S.C. performed surgery and FFN loading for in vivo experiments. M.D. characterized in vivo release. Viral delivery and preparation of ChR2-expressing animals was performed by K.A.K in E.R.K's laboratory. Manuscript was prepared by M.D. with significant contributions by D. Sames and D. Sulzer. All authors contributed to the design of the study and discussion of results.

## Additional information

**Competing interests:** The authors declare no competing interests.

