## [Peer Review File · Nature Communications]

Reviewers' comments:

Reviewer #1 (Remarks to the Author):

In this study, the authors extend the false neurotransmitter approach they pioneered into optical detection of noradrenergic (also called norepinephrine, NE) transmission in the CNS. The experiments are overall convincing and the data supports the specificity of the FFN 270 compound to vesicles that carry out NE release. This compound appears to show specificity of the NE surface membrane transporter as well as the vMAT2 vesicular transporter. Although, I think this manuscript deserves publication (as it provides substantial technical and conceptual advance over prior approaches), I believe a few key issues need clarification.

The authors claim that there are two populations of NE terminals segregated by their release probability. However, this issue is only probed by sustained 10 Hz stimulation. Medium to high frequency stimulations are known to be affected by short-term plasticity and therefore not reflect the initial release probability from terminals (e.g. Waters J, Smith SJ. *J Physiol.* 2002). Therefore, the authors should probe the same terminals with 1 Hz or lower stimulation frequencies to uncover any inherent differences in initial release probability.

What is the extent of spontaneous NE release in this system (e.g. the 27.6% decrease seen in ChR2 negative terminals reported on page 13)? Is this gradual unstimulated decrease in fluorescence baseline due to spontaneous release? Does this affect the measurements? If this is genuine release, does it also happen in "silent" boutons?

Reviewer #2 (Remarks to the Author):

In this manuscript Dunn et al. have synthesized a new tracer for the visualization of synaptic norepinephrine release from single synapses in ex vivo and in vivo preparations. Their data suggests that akin to what the groups have observed with dopamine transmission, there is differential release of norepinephrine at individual varicosities and that amphetamine can cause vesicular release of transmitter from noradrenergic neurons. While this compound is novel, the data are more appropriate for a more specialized journal.

Major issues:

DA neurons projecting to cortical areas are well known to have low DAT expression which could alter how permeable FF270 is to DA neurons in general. Why wasn't FF270 tested in an area like dorsal striatum where there should be very little innervation by noradrenaline containing neurons as a negative control?

While the authors focus on the LC as the primary area for innervation in the CNS, the A1/A2 cells groups are well known to project to the forebrain and indeed, the densest noradrenergic innervation in the forebrain is the bed nucleus of the stria terminalis which is mainly innervated by A1/A2 (which also innervates other amygdalar structures, the hypothalamus and even the insular cortex, see Robertson et al. *Nature Neuroscience*, 2013.) It would be very interesting to see if the A1/A2 neurons are functioning in the same manner (differential release sites, silent synapses) as the LC (A6) neurons, and would increase the impact of the manuscript. While this would be difficult in vivo, this should be feasible in slice experiments.

Due to the confound with ChR2 and FF270 emission wavelengths, precluding measurements during optical excitation, it would be prudent to repeat these experiments using a red-shifted opsin.

Minor points:

"The synaptic hypothesis of learning – which posits that ongoing alteration of synaptic transmission underlies an organism's ability to learn and change behaviors – is gathering experimental support." This is a very odd sentence as there has been support for plasticity going back decades. I suggest revising this.

Reviewer #3 (Remarks to the Author):

Dunn et al., claim the first fluorescent norepinephrine (NE)-like probes, FFN270, by its selective transport activity to both NET and VMAT2. They showed that FFN270 can be used as a tracer of NE at the single synapse levels in vivo. Overall, the results are quite impressive, and the concept of using a specific plasma membrane transporter, NET, for the development of NE-specific probe is very interesting. Moreover, the application of FFN270 to reveal the mechanism of amphetamine (AMPH) to the re-distribution of NE by the two-photon imaging in vivo was also an interesting example to emphasize its novelty. Although the novelty of this paper is enough to publish in Nature Communications as the reported VMAT2 fluorescent probes may not be used as NE-like fluorescent probes in vivo, there are several major questions to the authors for making this paper more impressive to wide audiences.

Major points)

1. According to the slices and the in vivo images, there are other major structures strongly labeled with FFN270 (Figure 3-6). Although the author presumed that those are blood vessels, there was no supporting data for proving the non-specific stained structures.
2. If blood vessels stained with FFN270, why the probe stains vessels. Is there a strong expression of NET or other types of non-selective monoamine transporter located in the blood vessels like SLC29A4?
3. Figure 4I) there are FFN270 positive, but TH-GFP negative axons. What is the origin of these axons? Are those NET expressed axons or non-specific binding of FFN270?
4. Figure 7) the authors' claim was overestimated. Although the authors observed the faster release of FFN270 by i.p. injection of AMPH, there is a possibility that it is due to inhibition of reuptake of FFN270 during their NE release by AMPH. So, it may not be conclusive unless measurement of the reuptake kinetics by using a reuptake inhibitor while the imaging of FFN270 provided.
5. Please include FFN102 staining in Supplementary Figure 4 to show its preferential selectivity to the three types of membrane transporters. If there is no DAT preferential staining of FFN102 than NET, NE neurons may also be labeled with a higher concentration of FFN102 by its activity as a VMAT2 substrate as shown in Figure 2. What is the author's opinion?

Minor points)

1. Based on its NET transporting activity and its pKa value, the 093 probe may have good property as an FFN for NE. Did the authors test the probe in the acute slice as well? How was the result?
2. How fast FFN270 enter to the NET overexpressed cells? Will it happen within a few minutes? How much fluorescent intensity is changed if FFN270 and other probes incubated to NET-VMAT2 double overexpressed cells to control?
3. There is another type of monoamine transporter in plasma membrane (SLC29A4). Is the FFN270 or FFN102 act as a fluorescent substrate for this transporter?
4. The limitation of FFN202 for its usage as an NE probe seems to be due to its less hNET transporting activity as shown in Fig. 1F. However, the description of the line 133-134, especially "NET activity of FFN202 was confirmed using - (hNET-HEK)" make it difficult to understand. Please revise the sentences, and add the figure number for this information.
5. Figure 6) It is not clear that ChR- axon sustained its FFN270 fluorescent signal than ChR+ axon

after optogenetic stimulation. Please replace the D-F with clearer figures to serve as representatives for Fig 6C.

6. There is no data for the description of line 260-263, dealing the calcium-dependent stimulated vesicle exocytosis.

7. Line 292-294, please add figure number for the description of a result.

8. Kindly relocate the line of 323-327 to Discussion.

Reviewers' comments:

Reviewer #1 (Remarks to the Author):

In this study, the authors extend the false neurotransmitter approach they pioneered into optical detection of noradrenergic (also called norepinephrine, NE) transmission in the CNS. The experiments are overall convincing and the data supports the specificity of the FFN 270 compound to vesicles that carry out NE release. This compound appears to show specificity of the NE surface membrane transporter as well as the vMAT2 vesicular transporter. Although, I think this manuscript deserves publication (as it provides substantial technical and conceptual advance over prior approaches), I believe a few key issues need clarification.

We appreciate this reviewer's endorsement of our study.

1. The authors claim that there are two populations of NE terminals segregated by their release probability. However, this issue is only probed by sustained 10 Hz stimulation. Medium to high frequency stimulations are known to be affected by short-term plasticity and therefore not reflect the initial release probability from terminals (e.g. Waters J, Smith SJ. *J Physiol.* 2002). Therefore, the authors should probe the same terminals with 1 Hz or lower stimulation frequencies to uncover any inherent differences in initial release probability.

We agree about the importance of frequency-dependent plasticity and thus have conducted additional experiments using 1 Hz stimuli. We found that there is greater release heterogeneity of NE release sites in the cortex in response to 10 Hz than 1 Hz stimuli in this system.

In detail, we repeated optogenetic stimulus experiments in acute brain slices and compared the responses to 10 Hz, 2,400 pulses with 1 Hz, 240 pulses: thus, maintaining the same duration of stimulation (4 min). We chose 1 and 10 Hz stimulus as NE neuron tonic activity is ~1-3 Hz (Howells, F. M. et al. (2012) *Metab. Brain Dis.*), and burst firing is often modeled at 10 Hz (the protocol used in the original submission).

We found that the mean of the total fluorescence loss from NE axons is not different between 1 (55.5 ± 3.0%) and 10 Hz (65.4 ± 3.0%) ($p = 0.07$, Mann Whitney U test). This may seem surprising, but is similar to the responses of dopamine axons, which have a very slow rate of "refilling of the releasable pool" compared to excitatory synapses (see review by Sulzer, D. et al. (2016) *Basal Ganglia*).

However, we observed that the release heterogeneity was greater at 10 Hz ($p < 0.0001$) versus 1 Hz ($p = 0.2$) stimulation (D'Agostino and Pearson normality test). This contrasts with the results obtained in cultured hippocampal synapses (e.g., reference suggested by this reviewer, Waters J, Smith SJ. *J Physiol.* 2002) where greater differences in synaptic efficacy was revealed at a lower stimulation frequency. This is an interesting observation, but we have refrained from over-interpreting our results in the absence of a more detailed study of release properties. We now write on Page 14:

"We next examined the effect of stimulus frequency by measuring the amount of FFN released at 1 Hz (240 pulses) and 10 Hz (2400 pulses), using transgenic optogenetic tool delivery by crossing the TH-Cre line with one expressing ChR2 controlled by a loxP site (Ai32, see Methods). In the transgenic line, we measured significant release of FFN270 from ChR2+ axons at both 10 Hz (loss of fluorescence, 65.4 ±

3.0%) and 1 Hz ($55.5 \pm 3.0\%$, Supplemental Fig. S10, 2 slices per frequency per animal, 4 animals): these results were not statistically different from each other ($p = 0.07$, Mann-Whitney), or when compared to the 10 Hz condition using viral expression ($62.9 \pm 3.2\%$, $p = 0.05$, Kruskal-Wallis). Consistent with previous findings, the 10 Hz stimuli did not produce a normal distribution of puncta destaining (D'Agostino and Pearson test, $p < 0.0001$), however, the 1 Hz stimulation condition was better fit ($p = 0.2$)."

Supplementary Figure 10. Effect of optogenetic stimulation frequency on FFN270 release. Acute murine brain slices containing Layer 1-3 of the barrel cortex were collected from TH-Cre/Ai32 animals expressing ChR2-EYFP in monoaminergic neurons. A-B) Representative images of FFN270 before (A) and after (B) 4 min of 1 Hz 470 nm light stimulation (5 ms duration, 240 pulses). C-D) Representative images of FFN270 before (A) and after (B) 4 min of 10 Hz 470 nm light stimulation (5 ms duration, 2,400 pulses). E) Change in fluorescence of individual axons was quantified and then grouped depending on colocalization with the EYFP reporter. Average release following 10 Hz ($65.4 \pm 3.0\%$) and 1 Hz ($55.5 \pm 3.0\%$) were not significantly different each other (Mann-Whitney, $p = 0.07$) and comparable to previous observations (Fig. 6). Scale Bar: 20 μm .

2. What is the extent of spontaneous NE release in this system (e.g. the 27.6% decrease seen in ChR2 negative terminals reported on page 13)? Is this gradual unstimulated decrease in fluorescence baseline due to spontaneous release? Does this affect the measurements? If this is genuine release, does it also

happen in “silent” boutons?

We recently published a more thorough analysis of this issue using ChR-dependent NE release in the hippocampal slice, where we measured spontaneous release by HPLC.⁵⁶ In that study, we report that ~26% of extracellular NE (4.75/18.25 pmol/mg) was present without ChR stimulus.

The most obvious potential mechanisms for this release are synaptic vesicle fusion or reverse transport through NET. We are confident that reverse transport is not playing an important role, as perfusion of the slice with the NET inhibitor nomifensine, which is well established to block amphetamine mediated reverse transport, did not affect spontaneous FFN release. Our results from this experiment have been added to Supplemental Fig. S7.

If vesicular fusion is responsible, it must be “spontaneous” release, as we quantified the FFN rundown found in the cadmium inhibited slices (which blocks voltage-dependent Ca²⁺ channels), and observed a comparable rate of 21% loss over a similar time course.

The release of stimulus-independent FFN270 likewise appears to occur spontaneously from both silent and active axons, as can be seen in Fig. 5. The rundown of signal in ChR+ “low releasing” axons is 25% and in ChR- axons is 27%. This is also comparable to the rundown observed *in vivo* in the optogenetic experiment (18%) and the amphetamine injection experiment (24%).

Additionally, with FFNs, and fluorescent probes in general, some amount of photobleaching is expected depending on the imaging conditions: in our hands, this often ranges from a 5-10% decrease during similar exposure windows. Moreover, pH changes within the synapse or deterioration of the health of the tissue are also expected to affect FFN signal. These forms of fluorescence loss are accounted for in our conclusions by maintaining consistent imaging conditions between groups, and alternating the order of brain slices used per group to prevent health related artifacts.

We now write on Page 15:

“As a control, perfusion of the NET inhibitor (nomifensine, 2 μM) on a slice preloaded with FFN270 had no effect on the rate of FFN signal decay (Supplemental Fig. S7, two-way ANOVA, $p = 0.95$), suggesting very low spontaneous FFN reverse transport and reuptake in the acute brain slice.”

Reviewer #2 (Remarks to the Author):

In this manuscript Dunn et al. have synthesized a new tracer for the visualization of synaptic norepinephrine release from single synapses in ex vivo and in vivo preparations. Their data suggests that akin to what the groups have observed with dopamine transmission, there is differential release of norepinephrine at individual varicosities and that amphetamine can cause vesicular release of transmitter from noradrenergic neurons. While this compound is novel, the data are more appropriate for a more specialized journal.

We thank the reviewer for acknowledging the novel nature of the compound. In contrast to the other reviewers, there is a note that the data acquired with the new probe belongs to a more specialized journal. We kindly ask the reviewer to consider that the development of an NE FFN capable of providing

a useful level of signal in the living brain is not a trivial matter (e.g., design of a dual NET-VMAT2 fluorescent substrate, NET selective over DAT/SERT, clean pharmacological profile, no toxicity, and low background staining in brain tissue). Further, this new probe/technique already provided new data on presynaptic variability and pharmacological insights that could not be made previously, in the context of NE systems that are widely important in neuroscience. Importantly, to our knowledge, this is the first presynaptic optical measure of monoaminergic neurotransmission in brain *in vivo*.

Please also note that a new paper that independently, but less directly, confirms our discovery that most catecholamine synapses are apparently silent, was published this month by the Kaeser lab in *Cell* (Liu et al., 2018, Dopamine secretion is mediated by sparse active zone-like release sites), indicating that this is a timely topic with appeal beyond a small group of specialists.

Major issues:

1. DA neurons projecting to cortical areas are well known to have low DAT expression which could alter how permeable FF270 is to DA neurons in general. Why wasn't FF270 tested in an area like dorsal striatum where there should be very little innervation by noradrenaline containing neurons as a negative control?

We thank the reviewer for this suggestion, and now compare the loading of FFN102 and FFN270 in both the dorsal striatum and barrel cortex, directly in pairs of acute brain slices from animals expressing tdTomato under the DAT-promoter (DAT-IRES-Cre/Ai9 cross). We used a high concentration of the probes (5X the optimal loading concentration, 50 μ M) to make sure we did not miss any concentration-dependent signal.

The results of these experiments are included in new Supplemental Fig. S9, and contribute three important new points to the manuscript:

- 1) There is no significant DAT-tdTomato label in neurites in Layers 1-3 of the barrel cortex, consistent with a lack or very low level of ventral midbrain dopamine innervation.
- 2) Even at 50 μ M, the DAT substrate FFN102 does not label axonal structures in the barrel cortex.
- 3) Even at 50 μ M, there is very little FFN270 label of neurites in the DAT-rich dorsal striatum (while the DA FFN probe, FFN102, gives strong labeling of DA axons).

We now write on Page 12:

“This conclusion is further supported by an absence of fluorescent reporter found in this region in a mouse line expressing tdTomato under a DAT-Cre driven promoter (see Methods), as well as the lack of substantial FFN270 uptake in a brain region rich in DAT expression, the dorsal striatum (Supplemental Fig. S9).”

Supplementary Figure 9. Comparison of FFN102 and FFN270 loading in dorsal striatum and barrel cortex. A-B) Acute slices containing the dorsal striatum (DS) were loaded with a high incubation dose (50 μ M) of either FFN102 (green) or FFN270 (magenta), and colocalization with the tdTomato reporter (red) in DAT-positive neurons was determined (see Methods for details). We observed much stronger FFN102 signal in the DS compared to FFN270, even at this elevated dose. C-D) This comparison was repeated in the barrel cortex, where we observed almost no dopamine axons (small scattered spots are likely

autofluorescence). In this brain region, we observed insignificant FFN102 uptake compared to FFN270, even with 50 μM incubation. Scale bar: 20 μm .

2. While the authors focus on the LC as the primary area for innervation in the CNS, the A1/A2 cells groups are well known to project to the forebrain and indeed, the densest noradrenergic innervation in the forebrain is the bed nucleus of the stria terminalis which is mainly innervated by A1/A2 (which also innervates other amygdalar structures, the hypothalamus and even the insular cortex, see Robertson et al. Nature Neuroscience, 2013.) It would be very interesting to see if the A1/A2 neurons are functioning in the same manner (differential release sites, silent synapses) as the LC (A6) neurons, and would increase the impact of the manuscript. While this would be difficult *in vivo*, this should be feasible in slice experiments.

In response to this suggestion, we now show labeling of NE axons with FFN270 in the bed nucleus of the stria terminalis, in Supplemental Figure S11. In this area, we observed a generally higher density of FFN270 positive axons, although the background signal was not as “clean” as in the cortex.

We agree that studying the differences between presynaptic release properties at additional NE synapses in the nervous system is valuable, but we believe that detailed examination of release properties of these sites is beyond the scope of this study. We elected to concentrate on NE in the barrel cortex as this is a site that 1) can be examined with a cranial window *in vivo* without highly invasive surgery and 2) NE plays important roles here that can be studied in rodents (e.g., whisker response learning paradigms).

Nevertheless, in the revised manuscript, we show FFN270 uptake in three dispersed NE-innervated regions in the nervous system (barrel cortex, bed nucleus, and LC), and hope that we provide a solid foundation for other researchers to use this probe to further characterize NE synapses throughout the CNS and PNS.

Supplementary Figure 11. FFN270 Loading in Bed Nuclei of the Stria Terminalis. A) Atlas image highlighting in red the location of the bed nuclei of the stria terminalis (BST) in the mouse brain (Bregma: -0.5 mm, Allen Institute).⁵¹ B) FFN270 (10 μM) labeling pattern in the BST following 30 min incubation. Scale Bar: 10 μm .

3. Due to the confound with ChR2 and FF270 emission wavelengths, precluding measurements during optical excitation, it would be prudent to repeat these experiments using a red-shifted opsin.

We agree that the red shifted ChRs, such as ChrimsonR, would be valuable to reduce the gap between stimulus and detection, but after significant work have had to settle for using only ChR2 in the cortex. Unfortunately, after injecting the TH-Cre animals in the LC with AAV5-hSyn-FLEX-ChrimsonR-tdTomato we were unable to observe any tdTomato in the distal NE projections in the cortex even after 8 weeks. This stands in contrast to responses with AAV/2/5.EF1a.DIO.hChR2(H134R)-EYFP.WPRE.hGH virus, where we observed robust EYFP signal after 4 weeks post-injection. We believe that improved expression or distal imaging techniques will be required for this objective.

To acknowledge the value of this potential future direction, we have added a sentence in the Discussion on Page 18.

“Alternative stimulation techniques, such as red-shifted ChRs, will enable more detailed examination of release properties of NE release sites by measuring FFN release during optogenetic stimulation of NE axons (local stimulation) or somata (distal stimulation).”

Minor points:

4. “The synaptic hypothesis of learning – which posits that ongoing alteration of synaptic transmission underlies an organism’s ability to learn and change behaviors – is gathering experimental support.” This is a very odd sentence as there has been support for plasticity going back decades. I suggest revising this.

This is a good point, and we have changed this sentence accordingly on Page 4.

“The ongoing alteration of synaptic transmission provides a means for the organism’s ability to learn and change behaviors.”

Reviewer #3 (Remarks to the Author):

Dunn et al., claim the first fluorescent norepinephrine (NE)-like probes, FFN270, by its selective transport activity to both NET and VMAT2. They showed that FFN270 can be used as a tracer of NE at the single synapse levels in vivo. Overall, the results are quite impressive, and the concept of using a specific plasma membrane transporter, NET, for the development of NE-specific probe is very interesting. Moreover, the application of FFN270 to reveal the mechanism of amphetamine (AMPH) to the re-distribution of NE by the two-photon imaging in vivo was also an interesting example to emphasize its novelty. Although the novelty of this paper is enough to publish in Nature Communications as the reported VMAT2 fluorescent probes may not be used as NE-like fluorescent probes in vivo, there are several major questions to the authors for making this paper more impressive to wide audiences.

We thank the reviewer for the endorsement of our new study and helpful critiques.

Major points)

1. According to the slices and the in vivo images, there are other major structures strongly labeled with FFN270 (Figure 3-6). Although the author presumed that those are blood vessels, there was no supporting data for proving the non-specific stained structures.

In this revision, we confirm that these structures are blood vessels using an established lectin stain (Bucher et al. 2014. *J. Cereb. Blood Flow Metab.* 34, 1128–1137) (procedure is described in the Methods on Page 30). The images are included in a new Supplemental Fig. S8, and discussed briefly on Page 11.

“After FFN270 incubation in acute brain slice, we observed significant labeling of noradrenergic axons (Fig. 4B), as well as larger structures that we confirmed were blood vessels by lectin staining (Supplementary Fig. S8).”

Supplementary Figure 8. Colocalization of FFN270 with blood vessels. A-B) Two representative pairs of low zoom images highlighting colocalization between FFN270 (magenta) and lectin stained vasculature (green) across Layers 2-6 of the barrel cortex. Note that there is not perfect overlap between channels as the FFN270 images were collected in healthy acute brain slices, and the lectin stain was performed on those same slices post-fixation. Some clearer examples have been highlighted by red arrows. Scale bar: 100 μm .

2. If blood vessels stained with FFN270, why the probe stains vessels. Is there a strong expression of NET or other types of non-selective monoamine transporter located in the blood vessels like SLC29A4?

This important question is broadly pertinent for many fluorescent probes, which are often seen in blood vessels. We agree that additional transporters might be involved, as may binding to additional receptors and molecules.

First, regarding a role for transporters, FFN270 uptake into the blood vessels is NET, DAT, and SERT-independent, as there is unchanged uptake after nomifensine and reboxetine (Fig. 4), as well as imipramine (not shown). We now write on Page 12:

“However, the uptake in blood vessels was unchanged for both inhibition conditions.”

To identify additional potential binding sites of FFN270 incubation, we sent the probe to the Psychoactive Drugs Screening Program (PDSP), which is supported by NIMH. The results are shown below, and succinctly, detect no significant inhibition (> 50%) by FFN270 at 54 different potential targets. These results are now included in an additional Supplemental Table S2. We also have discussed these results in a new paragraph in the Results on Page 10:

“FFN270 does not bind to monoamine receptors and other CNS targets

A primary screening assay was conducted with FFN270 against 54 CNS molecular targets, including the monoamine receptors and transporters (in collaboration with the Psychoactive Drug Screening Program, University of North Carolina at Chapel Hill).⁴⁸ The screen showed no significant binding of FFN270 to any of the receptors examined (a positive hit is defined as >50% inhibition by the experimental ligand at 10 μ M, Supplementary Table 2). As with any compound used in pharmacological studies, there remain other potential targets including orphan receptors which cannot be excluded, but this primary assay indicated a clean pharmacological profile for FFN270, an important prerequisite for the FFN probe design.”

5-HT1A	5.7	Beta1	-17.3
5-HT1B	37.3	Beta2	-7.9
5-HT1D	-4	Beta3	-2.7
5-ht1e	11.5	D1	-2.5
5-HT2A	6.3	D2	-3.5
5-HT2B	3.5	D3	3.7
5-HT2C	2.5	D4	20.7
5-HT3	-8.5	D5	10.7
5-HT4	-2.8	DAT	34.2
5-ht5a	2.6	DOR	6.7

5-HT6	-4.4	GABAA	10.5
5-HT7	2.5	H1	5.6
A2B2	-3.4	H2	14.6
A2B4	5.5	H3	13.6
A3B2	-8.6	H4	2.8
A3B4	5.1	HERG binding	-4
A4B2	-3.6	KOR	7.3
A4B2	3.8	M1	5.6
A4B4	4.6	M2	-0.6
A7	-28.4	M3	-12.2
A7	6	M4	10.1
Alpha1A	-4.2	M5	1.5
Alpha1B	2.3	MOR	15.7
Alpha1D	-6.2	NET	14.3
Alpha2A	-2	SERT	-9
Alpha2B	-13.7	Sigma 1	-4.9
Alpha2C	-8.5	Sigma 2	2.2

Supplementary Table 2. Primary Screen of FFN270 binding at 54 CNS Receptors. FFN270 (10 μ M) was tested at 54 CNS receptors in collaboration with the Psychoactive Drug Screening Program (PDSP). Data represent mean % inhibition (N = 4 determinations) for FFN270 tested at receptor subtypes. Values greater than 50% are considered significant.

It is possible that additional monoamine transporters (such as SLC29A4) may be involved in FFN270's staining of blood vessels, however, identification of the responsible target(s) is beyond the scope of this study.

3. Figure 4I) there are FFN270 positive, but TH-GFP negative axons. What is the origin of these axons? Are those NET expressed axons or non-specific binding of FFN270?

The relatively small fraction of FFN270 signal that is not inhibited by nomifensine (7%), colocalized with TH-GFP (11%), or released with AMPH (9%) do not appear to be NET axons, and therefore can be considered to represent "nonspecific signal" of the probe. We now write on Page 15.

“The remaining 9.2% of puncta that did not destain after the entire time course likely represent a nonspecific signal of FFN270 in acute brain slices, and was comparable to the number of puncta that remained in slice after inhibition with nomifensine ($7.2 \pm 0.7\%$, Fig. 4F) and did not colocalize with the TH-GFP signal (11.1%, Fig 4I).”

4. Figure 7) the authors' claim was overestimated. Although the authors observed the faster release of FFN270 by i.p. injection of AMPH, there is a possibility that it is due to inhibition of reuptake of FFN270 during their NE release by AMPH. So, it may not be conclusive unless measurement of the reuptake kinetics by using a reuptake inhibitor while the imaging of FFN270 provided.

This is an excellent recommendation, and indeed our lab has previously published the kinetics of AMPH inhibition of DAT mediated uptake. We have now addressed this concern by performing an experiment in which we perfuse the labeled acute slice with nomifensine ($2 \mu\text{M}$), a well-established reuptake inhibitor of NET. We found that nomifensine had no significant effect on FFN270 signal or rundown kinetics (Supplemental Fig. S7). We now write on Page 15:

“As a control, perfusion of the NET inhibitor (nomifensine, $2 \mu\text{M}$) on a slice preloaded with FFN270 had no effect on the rate of FFN signal decay (Supplemental Fig. S7, two-way ANOVA, $p = 0.95$), suggesting very low spontaneous FFN reverse transport and reuptake in the acute brain slice. This control supports a model in which amphetamine-dependent release is due to a redistribution of catecholamines from synaptic vesicles to the cytosol from where it undergoes reverse transport,⁵⁸ and that in this system, inhibition of FFN re-uptake plays little role in amphetamine effects.”

Supplementary Figure 7. Amphetamine-induced FFN270 release in acute brain slice. A) Acute slices were loaded with FFN270 and then imaged every 1 min. The average change in fluorescence of FFN270 puncta was measured during the course of an amphetamine (AMPH, 10 μM), nomifensine (Nom., 2 μM), or ACSF perfusion (starting at t = 0). B) the change in number of selected FFN270 puncta before and 5 min after AMPH or Nom. perfusion. Representative FFN270 images before and 5 min after ACSF (C-D), Nom. (E-F), or AMPH (G-H) perfusion. Scale bar: 10 μm.

5. Please include FFN102 staining in Supplementary Figure 4 to show its preferential selectivity to the three types of membrane transporters. If there is no DAT preferential staining of FFN102 than NET, NE neurons may also be labeled with a higher concentration of FFN102 by its activity as a VMAT2 substrate as shown in Figure 2. What is the author's opinion?

We thank the reviewer for this thoughtful suggestion, and now compare the loading of FFN102 and FFN270 in both the dorsal striatum and barrel cortex, directly in pairs of acute brain slices from animals expressing tdTomato under the DAT-promoter (DAT-IRES-Cre/Ai9 cross). We further used a higher concentration of the probes (5X the optimal loading concentration, 50 μ M) to examine the selectivity at two different concentrations and to examine the possibility of VMAT2-driven labeling at higher concentrations.

The results of these experiments are included in new Supplemental Fig. S9, and contribute three important new points to the manuscript:

- 1) There is no significant DAT-tdTomato label in neurites in Layers 1-3 of the barrel cortex, consistent with a lack or very low level of ventral midbrain dopamine innervation.
- 2) Even at 50 μ M, the DAT substrate FFN102 does not label axonal structures in the barrel cortex.
- 3) Even at 50 μ M, there is very little FFN270 label of neurites in the DAT-rich dorsal striatum.

We now write on Page 12:

“This conclusion is further supported by an absence of fluorescent reporter found in this region in a mouse line expressing tdTomato under a DAT-Cre driven promoter (see Methods), as well as the lack of substantial FFN270 uptake in a brain region rich in DAT expression, the dorsal striatum (Supplemental Fig. S9).”

Supplementary Figure 9. Comparison of FFN102 and FFN270 loading in dorsal striatum and barrel cortex. A-B) Acute slices containing the dorsal striatum (DS) were loaded with a high incubation dose (50 μ M) of either FFN102 (green) or FFN270 (magenta), and colocalization with the tdTomato reporter (red) in DAT-positive neurons was determined (see Methods for details). We observed much stronger FFN102 signal in the DS compared to FFN270, even at this elevated dose. C-D) This comparison was repeated in the barrel cortex, where we observed almost no dopamine axons (small scattered spots are likely autofluorescence). In this brain region, we observed insignificant FFN102 uptake compared to FFN270, even with 50 μ M incubation. Scale bar: 20 μ m.

Minor points)

1. Based on its NET transporting activity and its pKa value, the 093 probe may have good property as an FFN for NE. Did the authors test the probe in the acute slice as well? How was the result?

Prior to synthesis of the complete series of 201 derivatives, **093** was explored, but it did not perform as well as FFN270. We have included an extra representative image of **093** in acute brain slice in Supplemental Figure S5 and brief mention on Page 11.

“FFN202 and **093** showed low and moderate levels of uptake, respectively (Supplementary Fig. S5).”

Supplementary Figure 5. FFN202 and 093 staining Layer 1 of the barrel cortex in acute murine brain slice. Representative images of FFN202 (A) and **093** (B) in Layer 1 of the barrel cortex following a 30 min incubation (10 μ M). Highlighted with red arrows are labeled axon structures. Scale Bar: 10 μ m.

2. How fast FFN270 enter to the NET overexpressed cells? Will it happen within a few minutes? How much fluorescent intensity is changed if FFN270 and other probes incubated to NET-VMAT2 double overexpressed cells to control?

Uptake of **201** and FFN270 in NET-HEK cells is rapid (minutes), and we include below an example graph demonstrating the linear range of detectable uptake (Signal – Background) over 18 min. Note that detectable signal above background is observed by the first time point (3 min).

While in principle, combining the VMAT2 and NET assays should provide a better model representation of the presynaptic botouns in tissue, it is harder to interpret in terms of NET transport (e.g., a better VMAT2 substrate would be sequestered into acidic vesicles to a greater extent, resulting in lower fluorescence signal due to acidification and the inner filter effect). Our current evaluation strategy, where the compounds are examined first in cell lines expressing a single transporter, directly followed by studies in brain tissue, appears to provide sufficient throughput for the purposes of this study.

3. There is another type of monoamine transporter in plasma membrane (SLC29A4). Is the FFN270 or FFN102 act as a fluorescent substrate for this transporter?

We have not examined FFN270 as a potential substrate for uptake-2 transporters such as SLC29A4. Please note that this transporter is likely not a factor in the uptake of FFN270 by NE axons (as over 90% of FFN270 puncta uptake is dependent on NET activity). We agree that it, and additional transporters, are nevertheless good potential candidates for characterization of blood vessel staining. We believe that the characterization of blood vessel label is beyond the scope of this study, but represents an interesting avenue for future research.

4. The limitation of FFN202 for its usage as an NE probe seems to be due to its less hNET transporting activity as shown in Fig. 1F. However, the description of the line 133-134, especially “NET activity of FFN202 was confirmed using - (hNET-HEK)” make it difficult to understand. Please revise the sentences, and add the figure number for this information.

This line on Page 7 has been changed to make the statement clearer:

“Consistently, we observed a small level of accumulation of FFN202 in human embryonic kidney cells stably transfected with human NET (hNET-HEK) (Fig. 1F).”

5. Figure 6) It is not clear that ChR- axon sustained its FFN270 fluorescent signal than ChR+ axon after optogenetic stimulation. Please replace the D-F with clearer figures to serve as representatives for Fig 6C.

Thank you for the suggestion, and we have now changed the arrows on the figure to highlight the difference between FFN270-positive axons that do (blue) or do not (red) have ChR2.

Figure 6. Examining NE axons and release sites in living animals. A) Representative *in vivo* setup using an anesthetized head-fixed animal to image Layer 1 of the barrel cortex. B) A representative 3-D reconstruction of FFN270 labeling (50 μ M locally applied) in Layer 1 of the barrel cortex *in vivo*. Scale Bar: 20 μ m. C). FFN270 loaded into NE axons with ChR2, as described in Figure 5, can be released with local 470 nm light stimulation (10 Hz, 2,400 pulses). There is a significant increase in FFN270 released from axons that colocalize with ChR2 (ChR+, 46.2 \pm 8.3%, highlighted with blue arrows) compared to axons that do not (ChR-, 18.13 \pm 4.8%, highlighted with a red arrow, n = 6 different animals, p = 0.006). Representative images of the ChR2-YFP signal (D), and the FFN270 signal before (E) and after (F) optogenetic stimulation. Scale Bar: 10 μ m.

6. There is no data for the description of line 260-263, dealing the calcium-dependent stimulated vesicle exocytosis.

We have included the puncta selection input and output of the Matlab script on Page 12.

“Repeating the electrical stimulation while inhibiting calcium channels using Cd²⁺ (200 μ M) led to an 87% reduction in the number of identified destaining puncta (2 out of 143 puncta vs. 17 out of 155), confirming that FFN270 release was due to calcium-dependent stimulated exocytosis (2-3 slices per animal, 3-4 animals).”

7. Line 292-294, please add figure number for the description of a result.

“Fig. 4F” has been added after this result.

8. Kindly relocate the line of 323-327 to Discussion.

This sentence has been removed from the Results, and partially reincorporated into the Discussion on Page 18.

“Alternative stimulation techniques, such as red-shifted ChRs, will enable more detailed examination of release properties of NE release sites by measuring FFN release during optogenetic stimulation of NE axons (local stimulation) or somata (distal stimulation).”

REVIEWERS' COMMENTS:

Reviewer #1 (Remarks to the Author):

The authors have diligently addressed my earlier questions. I do not see any issues that preclude publication. These results are quite timely and important.

Reviewer #2 (Remarks to the Author):

While, as the authors note, I am not as enthusiastic as the other two reviewers, I do concede that the additional experiments performed by the authors have improved the quality of the manuscript. Further, the authors have reasonably argued for the novelty of their manuscript. A minor issue that remains is that the data examining the effect of optogenetic stimulation frequency on the FFN270 release (Sup. figure 10) is tantalizingly close to showing significant differences between the groups (1 and 10 Hz stimulation). Is this due to a lack of statistical power with only 2 slices from 4 animals?

Reviewer #3 (Remarks to the Author):

The manuscript described the first fluorescent false neurotransmitter based on coumarin, which can be selectively transported by NET and VMAT2. Though the revised paper has been improved a lot, there is still room for improvements.

1. Data for FFN102 lead identification of NET is missing.
2. The authors claimed that Cd²⁺ (200 μM) led to an 87% reduction in destaining puncta. To support this point, supporting images are required.

REVIEWERS' COMMENTS:

Reviewer #1 (Remarks to the Author):

The authors have diligently addressed my earlier questions. I do not see any issues that preclude publication. These results are quite timely and important.

We thank the Reviewer for their endorsement and helpful comments during the review process.

Reviewer #2 (Remarks to the Author):

While, as the authors note, I am not as enthusiastic as the other two reviewers, I do concede that the additional experiments performed by the authors have improved the quality of the manuscript. Further, the authors have reasonably argued for the novelty of their manuscript. A minor issue that remains is that the data examining the effect of optogenetic stimulation frequency on the FFN270 release (Sup. figure 10) is tantalizingly close to showing significant differences between the groups (1 and 10 Hz stimulation). Is this due to a lack of statistical power with only 2 slices from 4 animals?

To address this point, we have changed the text in the Supplemental Fig. 10 legend to make clearer that this data was collected from 2 slices per condition per animal and from 4 different animals. Therefore, for each stimulation frequency, the data was collected from 8 independent runs. This resulted in a comparable number of total axons to the original data as presented in Figure 5, and so was consistent with the other experiments.

“Supplementary Figure 10. Effect of optogenetic stimulation frequency on FFN270 release. Acute murine brain slices containing Layer 1-3 of the barrel cortex were collected from TH-Cre/Ai32 animals expressing ChR2-EYFP in monoaminergic neurons. A-B) Representative images of FFN270 before (A) and after (B) 4 min of 1 Hz 470 nm light stimulation (5 ms duration, 240 pulses). C-D) Representative images of FFN270 before (A) and after (B) 4 min of 10 Hz 470 nm light stimulation (5 ms duration, 2,400 pulses). E) Change in fluorescence of individual axons was quantified and then grouped depending on colocalization with the EYFP reporter. Average release \pm SEM following 10 Hz ($65.4 \pm 3.0\%$) and 1 Hz ($55.5 \pm 3.0\%$) were not significantly different each other (Mann-Whitney, $P = 0.07$, $n = 2$ slices per condition per animal, 4 animals) and comparable to previous observations (Fig. 6). Scale Bar: 20 μm .”

Reviewer #3 (Remarks to the Author):

The manuscript described the first fluorescent false neurotransmitter based on coumarin, which can be selectively transported by NET and VMAT2. Though the revised paper has been improved a lot, there is still room for improvements.

1. Data for FFN102 lead identification of NET is missing.
2. The authors claimed that Cd^{2+} (200 μM) led to an 87% reduction in destaining puncta. To support this point, supporting images are required.

In our preliminary *in vivo* experiment we examined both FFN102 and FFN202, and in the prior version, included only the data from FFN202 in Supplemental Figure 2. We have now updated this to include a representative 2D image indicating that we did not observe any axonal staining in this experiment. This is consistent with the results obtained in acute murine brain slices (Figure 4 and Supplemental Figure 9).

Supplementary Figure 2. FFN202 and FFN102 *in vivo*. Structures of FFN202 (A) and FFN102 (E), the first FFNs to be tested *in vivo* (Wenbio Gan, NYU). Two-photon fluorescence images of FFN202 (B) and FFN102 (D) taken in Layer 1 of the somatosensory cortex. Red arrow: cell body, Blue arrow: axonal structure. C) 3-D reconstruction of FFN202 labeling in the outermost 100 μm of somatosensory cortex. More axonal structures are observed traveling in the z-plane (blue arrow).

To address the second point, we now include a set of representative images from the Cd2+ experiments. This data is included in a new Supplemental Figure 12, and is comparable to the original uninhibited set in Figure 5B.

Supplementary Figure 12. FFN270 Electrical Stimulation with Cadmium ions. A representative set of images highlighting the change in FFN270 fluorescence signal over the course of a 10 Hz electrical stimulus while blocking calcium channels with cadmium chloride (200 μ M). See Figure 5 as a comparison with control FFN270 fluorescent changes. Scale Bar: 5 μ m.